# Adaptation to sub-optimal hosts is a driver of viral diversification in the ocean

Hagay Enav[1,3], Shay Kirzner[1], Debbie Lindell[1], Yael Mandel-Gutfreund[1,2] & Oded Béjà [1]

Cyanophages of the *Myoviridae* family include generalist viruses capable of infecting a wide range of hosts including those from different cyanobacterial genera. While the influence of phages on host evolution has been studied previously, it is not known how the infection of distinct hosts influences the evolution of cyanophage populations. Here, using an experimental evolution approach, we investigated the adaptation of multiple cyanophage populations to distinct cyanobacterial hosts. We show that when infecting an "optimal" host, whose infection is the most efficient, phage populations accumulated only a few mutations. However, when infecting "sub-optimal" hosts, different mutations spread in the phage populations, leading to rapid diversification into distinct subpopulations. Based on our results, we propose a model demonstrating how shifts in microbial abundance, which lead to infection of "sub-optimal" hosts, act as a driver for rapid diversification of viral populations.

[1] Faculty of Biology, Technion-Israel Institute of Technology, 32000 Haifa, Israel. [2] Department of Computer Sciences, Technion-Israel Institute of Technology, 32000 Haifa, Israel. [3] Present address: Department of Microbiome Science, Max-Planck Institute for Developmental Biology, 72076 Tübingen, Germany. Correspondence and requests for materials should be addressed to H.E.(email: hagayenav@gmail.com) or to O.B. (email: beja@technion.ac.il)

Cyanobacteria of the genera *Prochlorococcus* and *Synechococcus* are the most abundant photosynthetic prokaryotes in oceanic environments, contributing greatly to primary production on a global scale[1–3]. Viruses infecting cyanobacteria (cyanophages) are considered to be a major cause of cyanobacterial mortality[4,5], therefore acting as a selective force on host populations[6,7]. Phages further promote cyanobacterial evolution, as they serve as agents for horizontal gene transfer[8]. Moreover, they are thought to play a role in biogeochemical cycling of dissolved and particulate organic matter[9,10] and to drive its diel release into the environment[11].

Cyanophages can be classified into one of three morphologically defined groups: *Podoviridae*, *Siphoviridae*, and *Myoviridae*[12–14]. While cyanophages of the first two groups are mostly host-specific, many cyanophages of the T4-like *Myoviridae* family (cyanomyophages) are generalist parasites, capable of infection of multiple cyanobacterial hosts[12], even of different genera[14,15]. However, it is likely that different hosts are infected with varying efficiency by the same cyanophage, as was demonstrated previously for phages infecting *Flavobacterium* and enteric hosts[16,17].

In natural habitats, the abundance of cyanobacterial types often changes over space and time[18–21], as a result of environmental changes or killing off of cyanobacterial hosts due to phage infection[22]. Moreover, host populations can evolve resistance to infections by a specific phage type, decreasing the number of potential hosts for a given phage[6,23]. Such changes in the abundance of host types that are infected most efficiently by a specific cyanophage can strongly influence the ability of the phage to reproduce. Thus, to maintain their reproduction, specific phage lineages would need to adapt to new hosts in their environment or else face the possibility of becoming extinct.

Previous studies that focused on the reciprocal coevolution between hosts and phages have shown that this type of evolution results in rapid diversification[7,16,24]. A study that examined the one-sided evolution of a generalist phage, adapted to heat stress in two different enteric bacterial hosts, identified convergent evolution in phage populations[25]. However, the question of phage adaptation to optimal versus sub-optimal hosts has yet to be addressed in environmentally relevant systems.

## Results and Discussion

In this study, we investigated the adaptation of replicate cyanophage populations to different cyanobacterial hosts to reflect shifts in the abundance of hosts in natural environments. Using experimental evolution of cyanomyophage populations in three cyanobacterial hosts from two different genera, we investigated the extent to which the optimality of the host shapes phage evolution. In this context, optimality relates to the efficiency with which the phages infect their cyanobacterial hosts, measured by the rate of decline of the host population. This has been shown previously to be directly correlated with phage fitness using a heterotrophic host and an RNA phage[26] and a marine cyanobacterial host and a DNA cyanophage[27] (see Supplementary Figure 1). This measure encompasses multiple ways in which fitness can change, including rate of adsorption, the latent period (the time at which new phage progeny are released), and burst size. Our results show, at the genomic and phenotypic levels, that infection of sub-optimal hosts is a major driver of viral evolution. This suggests that interactions with different host types is constantly shaping the structure of phage populations in natural environments, where host ecotypes coexist and form dynamic communities.

To examine how generalist cyanophages adapt to optimal and sub-optimal hosts, we performed an evolutionary experiment whereby evolving phage populations were used to infect naïve cyanobacterial hosts. This was done for 15 rounds of serial infection. A single isolate of the generalist S-TIM4 myovirus[6] was used to infect three different cyanobacterial strains: *Prochlorococcus* sp. strain MIT9515, *Prochlorococcus* sp. strain MED4, and *Synechococcus* sp. strain WH8102 (referred to from now by genus and strain names). This was done with 4–5 replicate viral populations for each interaction with the different hosts. To avoid coevolutionary dynamics and maintain the selective forces imposed by the host constant throughout the experiment, remaining cells were removed from phage lysates prior to initiation of the next round of infection (Fig. 1).

Following the evolutionary experiment, we performed whole population genome sequencing, with two sequencing libraries per evolved population and used the average frequency of mutations identified in both technical repeats. Additionally, to test for changes in the infection efficiencies of each phage population on each host, we infected the three hosts with the ancestral and each of the evolved phage populations.

The S-TIM4 phage was isolated on *Synechococcus* WH8102[6] and carries 235 genes in a 176 kb genome. Infectivity tests of the ancestral (unevolved) S-TIM4 phage showed that it infects *Prochlorococcus* MIT9515 with the highest efficiency (Supplementary Table 1). Therefore we refer to this cyanobacterium as the "optimal" host. *Prochlorococcus* MED4 and *Synechococcus* WH8102 are infected with lower efficiency, and are therefore considered "sub-optimal" hosts, with *Synechococcus* WH8102 being infected least efficiently.

**Mutants in cyanophage populations are positively selected.** Overall, we sequenced 14 S-TIM4 populations and identified 151 mutations in 86 different genomic positions (Supplementary Table 2) that were localized in 20 protein-coding genes (Supplementary Table 3). Ninety percent of the detected mutations were single nucleotide polymorphisms (SNPs), out of which 89% were nonsynonymous (Fig. 2a). The fraction of nonsynonymous mutations within coding regions is expected to be ~75% in the absence of selection and <75% if negative selection is the dominant type of selection[28]. Therefore, the significantly larger fraction we observed of 89% suggests that the emergence of the identified mutations is mostly the result of strong positive selection (Fisher's exact test $P$ value = $3 \times 10^{-3}$). This is true even when considering that some of these mutations could result from genetic hitch-hiking and possibly be swept later[29]. An additional 12 synonymous SNPs modify viral codons, so that the mutated codons could form full Watson–Crick pairing with the tRNAs encoded in the host genome (Fig. 2a, discussed below). The latter provide further evidence supporting that the mutations identified in this study are the result of positive selection.

Examination of the number of mutations per population showed that S-TIM4 populations that evolved in the *Synechococcus* WH8102 host accumulated significantly more mutations, and in more genes, than populations evolved in the *Prochlorococcus* hosts (Fig. 2b). The greater accumulation of mutations in these populations could be a result of cellular mechanisms of *Synechococcus* WH8102, making phages that were passaged through this host more prone to mutations. To examine this possibility, we carried out an additional evolutionary experiment with cyanomyophage Syn19[30] evolved in the same *Synechoccus* WH8102 host strain for the same number of serial infection rounds. These populations accumulated only 4–6 mutations per population (Supplementary Table 5). Therefore, we conclude that the greater accumulation of mutations in S-TIM4 populations passaged through WH8102 was not caused by the host cell per se, but was a result of a facilitated evolutionary process resulting from the interaction between the phage and this sub-optimal

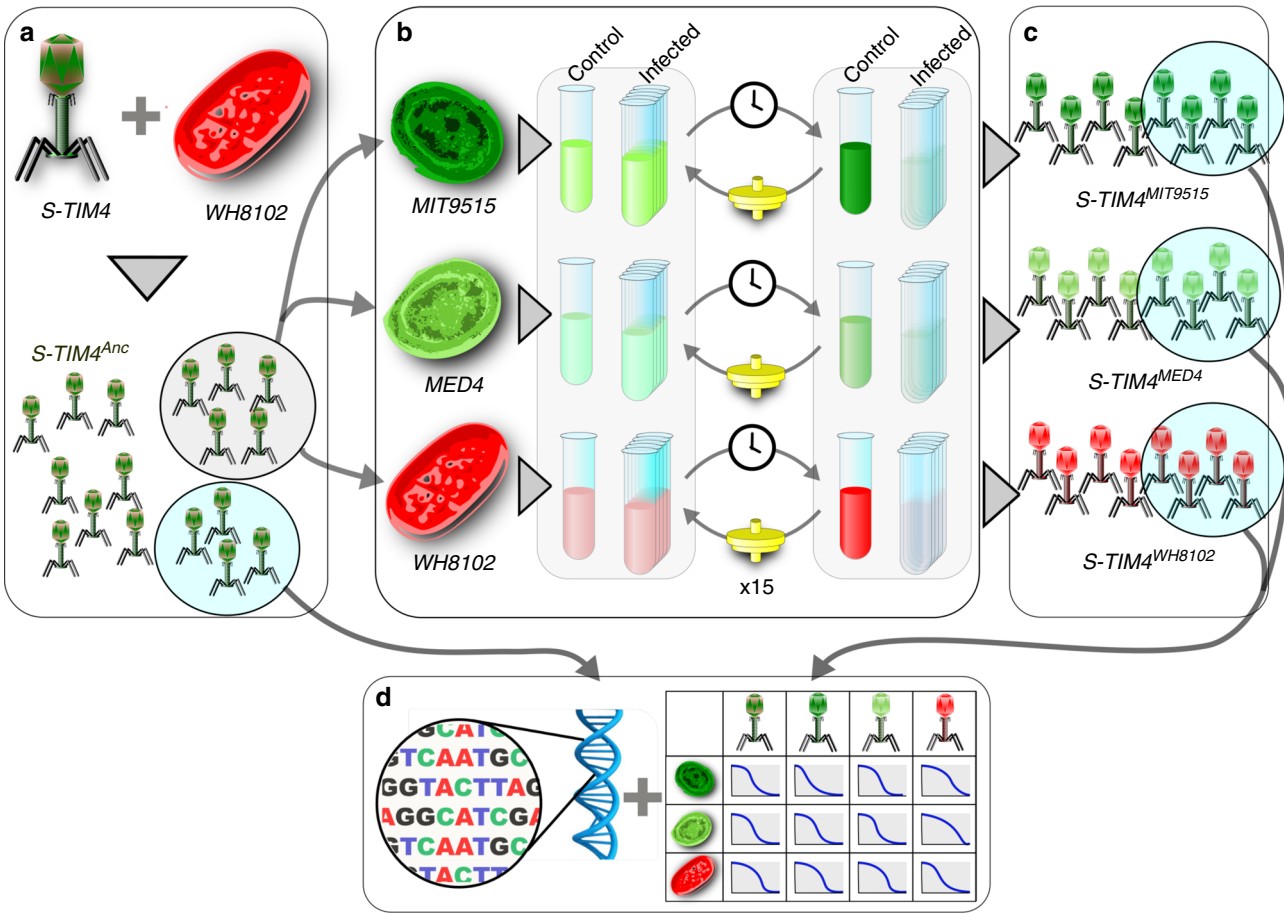

**Fig. 1** Procedure for the experimental evolution of cyanophage S-TIM4 populations. **a** Generation of the ancestral population. A single viral plaque was isolated and replicated on the *Synechococcus* sp. strain WH8102 host, to yield an ancestral phage population (S-TIM4^Anc). **b** Adaptation to specific hosts. Aliquots of the ancestral population were evolved in three distinct cyanobacterial hosts: *Prochlorococcus* sp. strain MIT9515 (dark green), *Prochlorococcus* sp. strain MED4 (light green), and *Synechococcus* sp. strain WH8102 (red). Each infection interaction was repeated five times (Infected) with one uninfected control culture (Control) for each cyanobacterial strain. Following lysis of the infected cultures, phage lysates (<0.22 μm) were transferred to infect the same naïve cyanobacterial hosts. This process was repeated for an additional 14 transfers. **c** Evolved populations. The adaptation phase resulted in populations adapted to MIT9515 (S-TIM4^MIT9515), MED4 (S-TIM4^MED4), and to WH8102 (S-TIM4^WH8102). **d** Characterization of ancestral and evolved populations. Whole genome DNA sequencing of the entire population for the ancestral and each of the evolved populations was conducted (left). In parallel the infectivity of each population on each of the hosts was tested (right)

host. Examination of the mutation types in Syn19 populations revealed that all were nonsynonymous SNPs, further emphasizing the role of positive selection in the adaptive process of cyanophage populations.

**Phages adapted to different hosts are phenotypically distinct**. Next, we sought to determine whether S-TIM4 populations that had evolved different genotypes had distinct phenotypes, as reflected by their infectivity profiles which are an indication of phage fitness[26] (Supplementary Figure 1). The infectivity of each phage population was determined on each of the three hosts at two virus particle per cell ratios of 0.1 and 3, as it has been demonstrated previously that the virus-host ratio influences infection efficiency[17]. We observed specialization of the evolved phage populations where populations evolved in *Synechococcus* WH8102 had improved infectivity of this host, compared to the ancestral population (Fig. 3a, b). Furthermore, this came at a cost of a reduction in the infectivity of the two *Prochlorococcus* hosts. The reduced infection efficiency was so extreme in two of these populations that the ability to infect the MED4 host was lost (Fig. 3a, b). In some cases, populations evolved in each of the *Prochlorococcus* hosts also evolved to infect that host better than

the phages evolved in the other *Prochlorococcus* host. At times, this was at the cost of a reduction in the infectivity of the *Synechococcus* WH8102 host. Thus, we found that phage specialization was manifested as both improved infection of the host used for evolution and decreased infection of the other host types.

To further investigate the phenotypic difference between populations of phages that were passaged through different hosts, we computed a phenotypic distance tree based on the infectivity profiles 10 days post infection, at virus particle per cell ratios of 0.1 and 3 (Fig. 3c). Evolved phage populations clustered into three distinct groups, corresponding to the host they were evolved in. This clustering indicates that evolution of viral populations in the same bacterial host resulted in the most similar phenotypic profiles. Populations evolved in the *Synechococcus* host were clustered together to form the group with the highest distance from the ancestral population. Since *Synechococcus* was the least optimal host for the ancestral phage, these findings emphasize the increased phenotypic effect of adaptation to sub-optimal hosts.

**Evolution in sub-optimal hosts increases genetic diversity**. Next we sought a better understanding of how the phenotypic distances between phage populations that have evolved in different

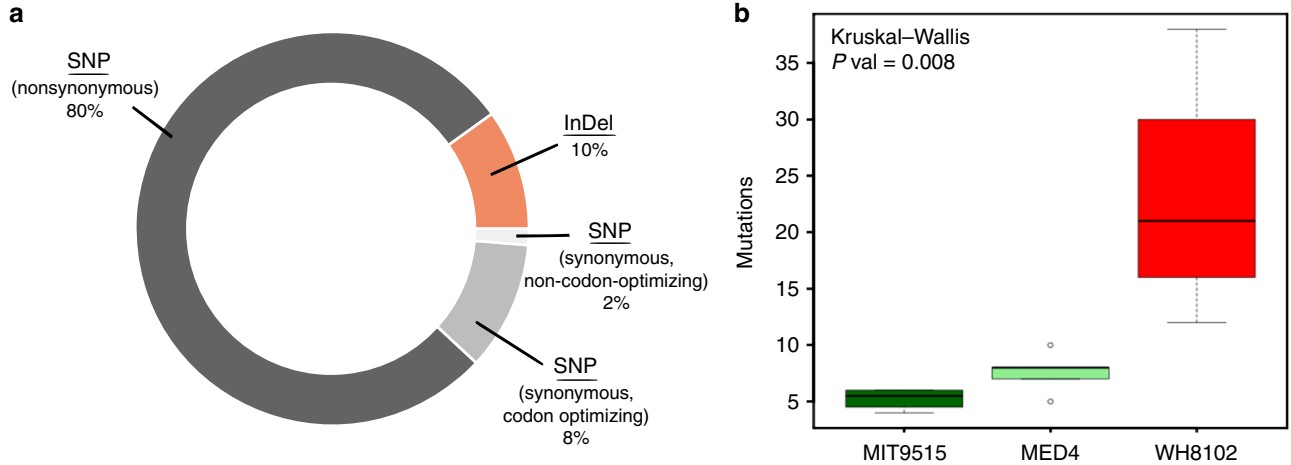

**Fig. 2** Mutations in evolved S-TIM4 populations are the result of strong positive selection. **a** Distribution of mutation types. Circle shows the distribution of mutations by type: insertions/deletions (orange), single nucleotide polymorphisms (SNPs) (gray scale: nonsynonymous—dark gray, synonymous codon-optimizing—medium gray, synonymous nonoptimizing codon—light gray). **b** Distribution of the number of mutations accumulated in evolved populations, grouped according to the host they evolved in. Populations evolved in WH8102 (red, $n = 4$) accumulated significantly more mutations, compared to populations evolved in either MIT9515 (dark green, $n = 4$) or MED4 (light green, $n = 5$) (Kruskal–Wallis $P$ value = 0.008). Raw data can be found in Supplementary Table 2

bacterial hosts are reflected in the mutational landscapes of the evolved populations. To do this, we created a genotypic profile for each phage population, containing the genomic positions and frequencies of all the mutations in the population (Fig. 3d). We then calculated the pair-wise genetic distances between each pair of phage populations and constructed a neighbor-joining tree (Fig. 3e). Phage populations were clearly grouped according to the bacterial host they evolved in, in a similar manner to the structure of the phenotypic distance tree. However, the genetic distances varied between phage populations that evolved in the same bacterial host. Distances among populations that evolved in the optimal host, *Prochlorococcus* MIT9515, were the lowest, while for populations evolved in the least optimal host, *Synechococcus* WH8102, the distances were highest (Fig. 3d, e).

Our combined findings suggest that strong purifying selection acted on mutations in S-TIM4 populations when evolving in the optimal MIT9515 hosts, with minimal genetic divergence occurring relative to the ancestral phage and the least diversity found among the different phage populations evolved in this host (Fig. 3e). However, when S-TIM4 populations evolved in the nonoptimal hosts, (i.e. *Synechococcus* WH8102 and *Prochlorococcus* MED4), the intensity of the purifying selection decreased, and positive selection resulted in the emergence of new diverse genotypes, with the greatest genetic divergence occurring during evolution on the least optimal host.

**Mutations in S-TIM4 are located in structural genes.** The majority of the 20 mutated S-TIM4 genes identified in this study are likely to be involved in building the viral particle. Based on homology to genes of the *Synechococcus* myophage Syn9[31], many are predicted to be expressed during the last phase of viral gene expression (late-expressed genes) (Fig. 4a and Supplementary Table 3), when most structural proteins are transcribed. Additionally, mass-spectrometry analysis of virus particles detected the proteins of 14 of the mutated genes (Supplementary Table 3), indicating that they are likely to have a structural role. Furthermore, all of the genes for these particle-associated proteins for which functions can be ascribed based on homology (eight genes) have structural functions[32] (Supplementary Table 3). All but two of them have a common structural function of being associated with the tail fibers and baseplate, which are responsible for host

recognition, attachment, and infectivity[32,33]. Populations evolved in the *Synechococcus* host had additional mutations in genes encoding components of the tail tube and in capsid formation (Fig. 4c). Of the six mutated genes that are not particle-associated, two have putative functional predictions of involvement in the response to nutrient limitation. These are the ORF23 (2OG[30]) and ORF224 (DUF680, also referred to as PhCOG173[34]) genes.

Mutations in four of the structural genes were common to the phage populations evolved in all three hosts (Figs. 3d, 4). In fact, five of the same six mutations that were identified in the phage populations evolved in *Prochlorococcus* MIT9515 were also found in nearly all of the other phage populations (Fig. 3d, Supplementary Table 2), further emphasizing the role played by positive selection. These findings suggest that these mutations emerged as an adaptation to a selective force imposed on all the cyanophage populations. This selective force could be a result of an intracellular component that is shared between hosts or possibly result from a feature of the extracellular environment (i.e. specific lab conditions) used in this study. This hypothesis also explains why populations evolved in the optimal host often have improved infectivity of sub-optimal hosts, compared to the ancestral phage (Fig. 3a, b).

The additional 16 mutated genes were found in phage populations that had evolved in either *Prochlorococcus* MED4 or *Synechococcus* WH8102, in sets of genes that were largely unique to each host (Fig. 4a, b). Among them only two genes were common to populations evolved in both of these hosts. This indicates that genetic adaptation was, for the most part, distinctly tailored to each of the sub-optimal hosts in which the phage populations were evolved.

Next we provide two substantially different examples of positive selection resulting in increased genetic diversity of phage populations during adaptation to sub-optimal hosts. The first is ORF108, encoding the YadA domain-containing structural protein, which was mutated in all populations, and the second is ORF224, encoding the DUF680-containing nonstructural protein (DUF, Domain Unknown function), that was mutated only in phages evolved in the *Synechococcus* WH8102 host.

The YadA domain-containing structural protein (2235 aa long) has a typical membrane adhesion-like domain that likely has a structural role in the assembly of the viral tail fibers that are

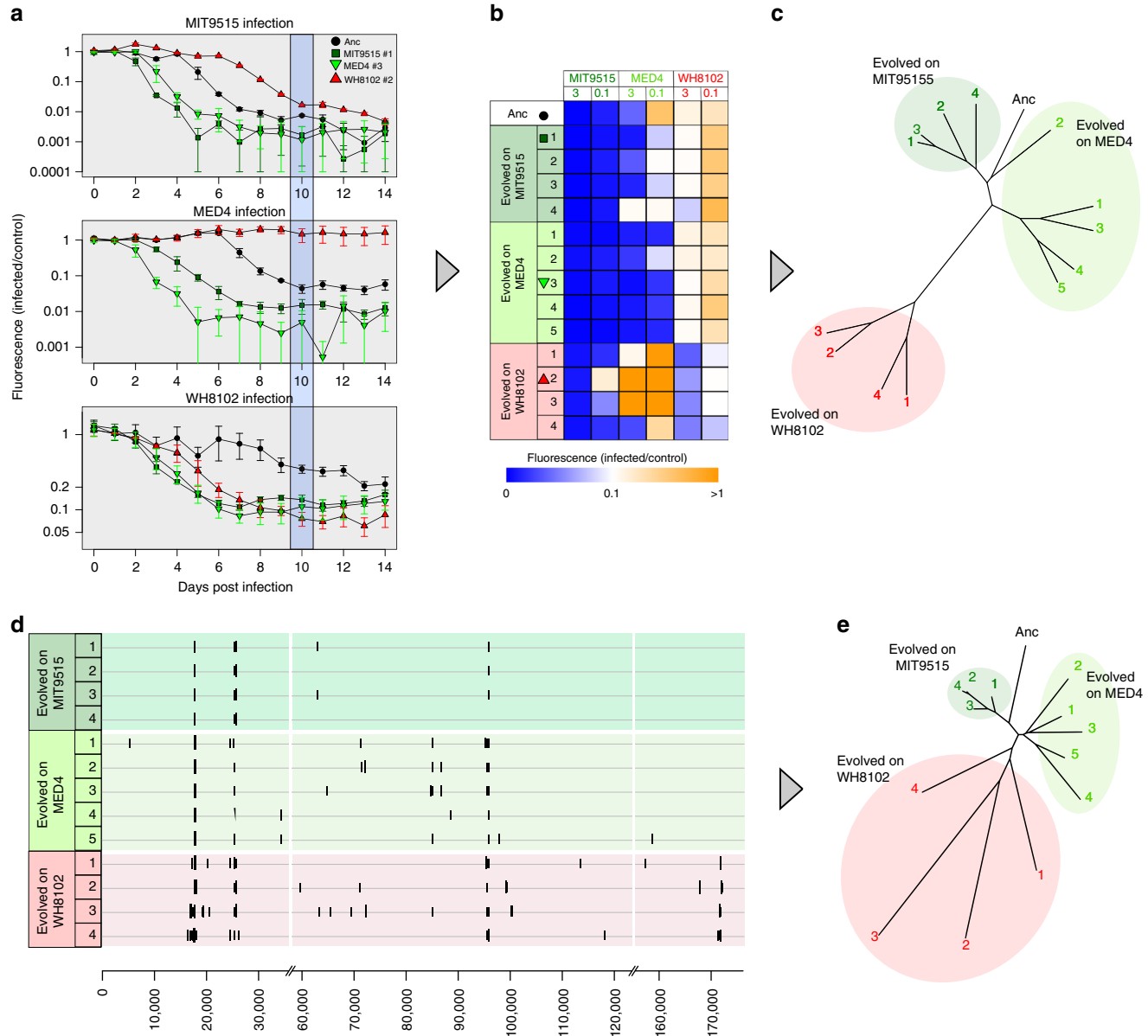

**Fig. 3** Phage populations adapted to the same host show high within-group similarity. **a** Representative normalized growth curves of the three hosts when infected by four phage populations at a virus particle per cell ratio of 3. Phage populations shown are the ancestral (Anc), evolved in the optimal host (MIT9515 #1), and evolved in sub-optimal hosts (MED4 #3, WH8102 #2). *Y*-axis shows the fluorescence levels of infected cultures divided by the fluorescence of the uninfected culture, i.e, normalized culture fluorescence. Blue shading shows normalized fluorescence on the tenth day post infection, which was used for further analyses. Each curve is the average of three infection assays, error bars correspond to standard deviations. **b** Normalized cell densities of all infected cyanobacterial cultures at 10 days post infection. Lower values indicate more rapid declines in the infected host population, i.e. increased fitness (Supplementary Figure 1). Rows correspond to viral populations, columns are for infections of specific hosts at 0.1 and 3 viral particles/ cell. Each box is the average of three infection assays. Symbols next to the population number denote the populations shown in panel **a**. **c** The phenotypic distance tree is based on the infection profile of each population (i.e., values of each row in panel **b**). Colors and numbering correspond to panel **b**. **d** Genomic maps show increased diversity in populations evolved in sub-optimal hosts. Positions along the genome are shown on the *x*-axis; horizontal lines correspond to specific populations. Vertical lines correspond to identified mutations, and their length corresponds to the mutation frequency within the population. **e** Genotypic distance tree reveals lower distance in populations evolved in MIT9515. The tree was constructed based on pair-wise genetic distances between phage populations. Colors and numbering correspond to panel **b**. Raw data can be found in Supplementary Table 2 and Supplementary File 1

involved in the initial phage attachment to the host cell[35,36]. This gene was mutated in a single position (Thr$_{580}$ = >Ala$_{580}$) and at similar frequencies (5−16%) in three of the four S-TIM4 populations passaged through *Prochlorococcus* MIT9515 as well as in populations evolved in the other two hosts. In contrast, mutations in the phage populations evolved in *Prochlorococcus* MED4 had ten mutations in four distinct sites in this gene. The

divergence was even higher in phage populations evolved in the *Synechococcus* WH8102 host, where we found 33 mutations in 26 different sites, 31 of them different to those in the populations evolved in *Prochlorococcus* MED4 (Fig. 4d). Strikingly, in one phage population passaged through *Synechococcus* WH8102 we identified 15 synonymous mutations that were restricted to a genomic region of 120 bp. While none of these mutations result

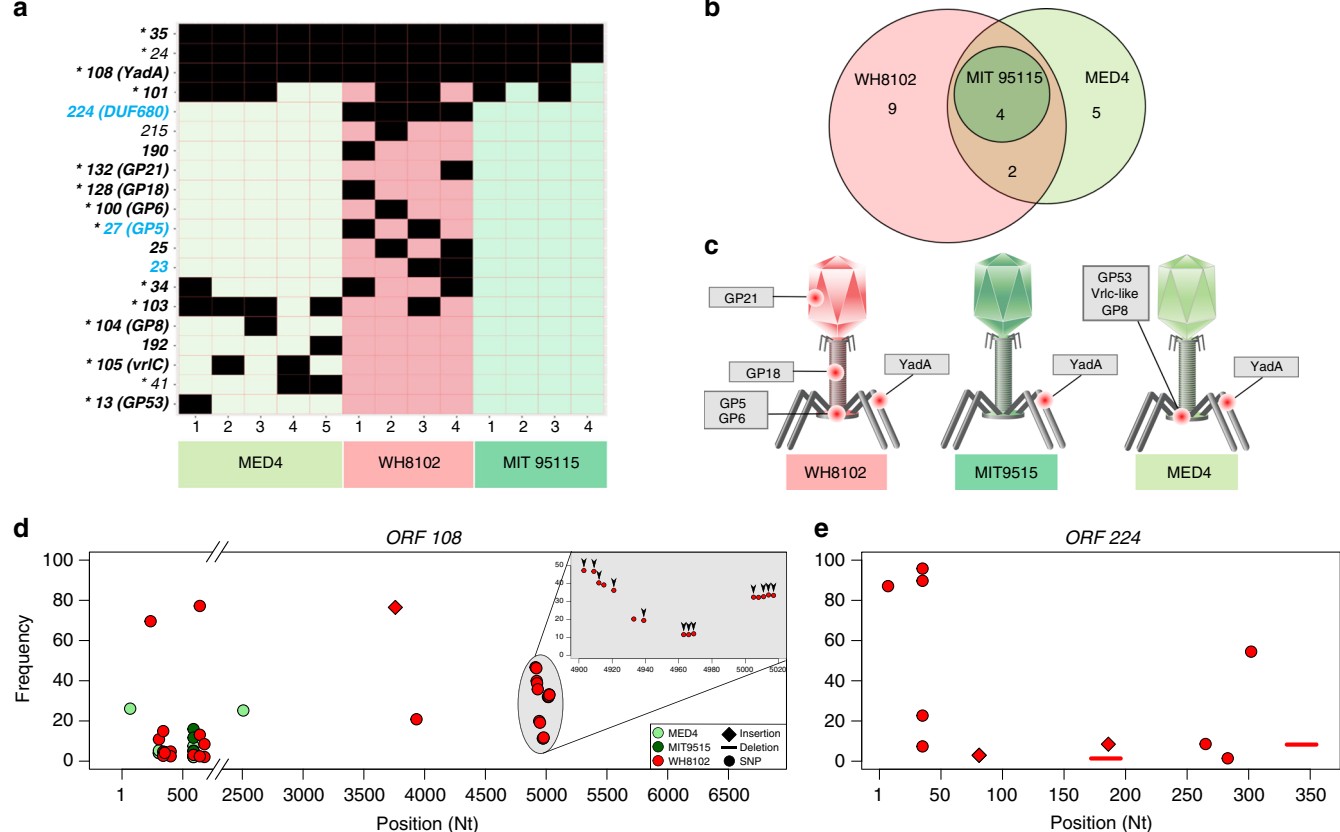

**Fig. 4** Discrete sets of structural genes are mutated in sub-optimal hosts. **a** Mutated genes in each evolved population. Each row represents one of the 20 mutated phage genes; columns represent evolved populations. Black cells indicate mutations identified in a particular gene. Genes in bold black type are predicted to be expressed during the late-phase of infection (and most have a structural role in the assembly of the virion particle). Genes in bold blue type are predicted to be middle-phase expressed genes. Genes marked with an asterisk encode proteins that were identified by mass spectrometry as part of the assembled viral particle. **b** Mutated genes in populations evolved in the optimal host are also mutated in other populations. Venn diagram showing the overlap of mutated genes in phage populations, grouped by the host they evolved in. **c** Location of mutated proteins in the virion particle. Each phage diagram represents populations evolved in a specific host: *Prochlorococcus* MIT9515, *Prochlorococcus* MED4, and *Synechococcus* WH8102. Protein names correspond to panel **a**. All proteins, excluding gp18 and gp21, are located in the tail fibers and baseplate, responsible for host recognition, attachment and infection. **d** Genetic map of ORF108 (YadA domain-containing structural protein) shows increased diversity and specificity of mutations in populations evolved in sub-optimal hosts. *X*-axis shows the positions along the gene, mutation frequencies are on the *Y*-axis. Gray inset plot is a zoom-in of a 120 base-pair long region which is enriched in 15 synonymous SNPs, all in the same population which evolved in WH8102 (#3). Black arrows are positioned over mutations that result in codon optimization to the host tRNA repertoire. **e** Genetic map of ORF224 (DUF680-containing protein) showing high diversity of mutations but only in populations evolved in *Synechococcus* WH8102. Raw data can be found in Supplementary Tables 2, 3

in adaptation to any of the three tRNA genes carried in the S-TIM4 genome, most mutations (12 out of 15) result in optimizing the codons to the host tRNA genes, thus potentially increasing the translation rate and accuracy of ORF108. These codon-optimizing SNPs were mostly found on the same sequencing reads, meaning that specific viruses carry clusters of these mutations (Supplementary Figure 2). A comparison of the mutation-rich region to the rest of the S-TIM4 genome revealed that the genomic region containing the ten most upstream mutations is identical to a region downstream to the mutated region, while the region containing the downstream mutations is identical to a region upstream to the mutated region. Therefore, we hypothesize that these polymorphisms are the result of at least two recombination events. Previously it was suggested that clusters of nonoptimal codons slow ribosome progression[37], possibly changing protein structure and function as a result of different folding dynamics[38,39]. In an earlier study we suggested that cyanomyophage genomes contain tRNA genes to allow improved translation of phage genes when infecting *Synechococcus* hosts[40]. The codon-optimizing mutations we identified in this study represent a different mechanism to overcome the codon

usage difference between cyanomyophages and their *Synechococcus* hosts.

The DUF680 protein (ORF224, also referred to as PhCOG173) is common in cyanomyophages[41] and was suggested to have a role in response to phosphate limitation[34]. This gene, which has no homologs in the bacterial strains used in this study, was mutated only in populations evolved in the *Synechococcus* host where it accumulated 13 different mutations, many of which are expected to interfere with its expression (i.e., frame-shifting indels, non-sense mutations and long deletions). Interestingly, in each population, the accumulative frequency of mutations in this gene is ~100%, suggesting that all phages in these populations carry a mutation in this gene. We speculate that the expression of the ancestral gene reduces the fitness of S-TIM4 when infecting *Synechococcus* WH8102 under the growth conditions used here. These two examples demonstrate starkly different means through which positive selection results in genetic diversification of phage populations, with the first likely improving the expression and functionality of the structural protein, while the second causes loss of function of a nutrient-response gene during adaptive evolution in nutrient-replete conditions.

**Variable genomic regions are not preferentially mutated.** Genes of cyanophages of the *Myoviridae* family can be divided into a conserved core-genome, which is present in all phages of that group, and to a flexible, horizontally transferred genome, which is expected to allow adaptation to specific environmental conditions[27,41,42]. We asked if phage adaptation to its hosts is achieved by preferential accumulation of mutations in either the core or flexible genomes using the previous classification of cyanomyophage genes, based on the comparison of 17 genomes[41]. We found that 8 of our 20 mutated genes belong to the core genome, i.e, they are included in all the cyanophage genomes of the *Myoviridae* group. Of the remaining 12 genes, nine appear in some cyanophage genomes while only three genes have no homologs in other phages (Fig. 5c).

Flexible genes often reside in hypervariable genomic regions[35]. Recently, it was suggested that genes responsible for host recognition and attachment reside within hypervariable genomic regions[27,35]. These regions are assumed to evolve rapidly to allow adaptation to new selective forces. This evolutionary adaptation occurs similarly in genomic islands described in bacterial genomes[43], often resulting in bacterial resistance to infection by specific phages[6]. We therefore sought to determine if the mutations we identified in the S-TIM4 genome are preferentially located in hypervariable metagenomics regions using the Global Ocean Sampling (GOS) metagenome dataset[44]. Interestingly, only 14.5% of the S-TIM4 mutations are in hypervariable regions, while 16.6% of the genome is defined as hypervariable (hypergeometric $P$ value = 0.728, Fig. 5a, Supplementary Tables 2,

4). Therefore, we conclude that phage adaptation to specific hosts is not preferentially mediated through mutations in these hypervariable genomic regions. It should be noted that in natural habitats, cyanophage diversification also occurs through allelic exchange between phages[45,46], which could not be investigated using our experimental system as only a single phage was present. Overall, these data demonstrate that phage adaptation to a specific host is not gained by exclusive modification to either the core or the flexible genome.

In a previous study, it was demonstrated that cyanophage ecotypes remain present in a natural habitat[45], possibly by rapid recombination events which lead to a stable persistence of the phage population and unstable association with the host ecotypes[47]. It was also suggested that discrete population boundaries are initiated by sympatric niche differentiation and maintained by recombination[46]. Based on our results, we support the latter; however, we propose that mutation and genetic drift play a key role in the initial formation of distinct phage populations.

**Model for the influence of host type on phage evolution.** Based on our results, we propose a model for the influence of bacterial host type on the evolution of phage populations (Fig. 6). According to our model, generalist cyanophages can infect a number of bacterial strains, with different degrees of efficiency, as was shown previously in other phage-host systems[16,17,48]. The bacterial strain infected with the highest efficiency is the "optimal" host, while other host strains are "sub-optimal". When

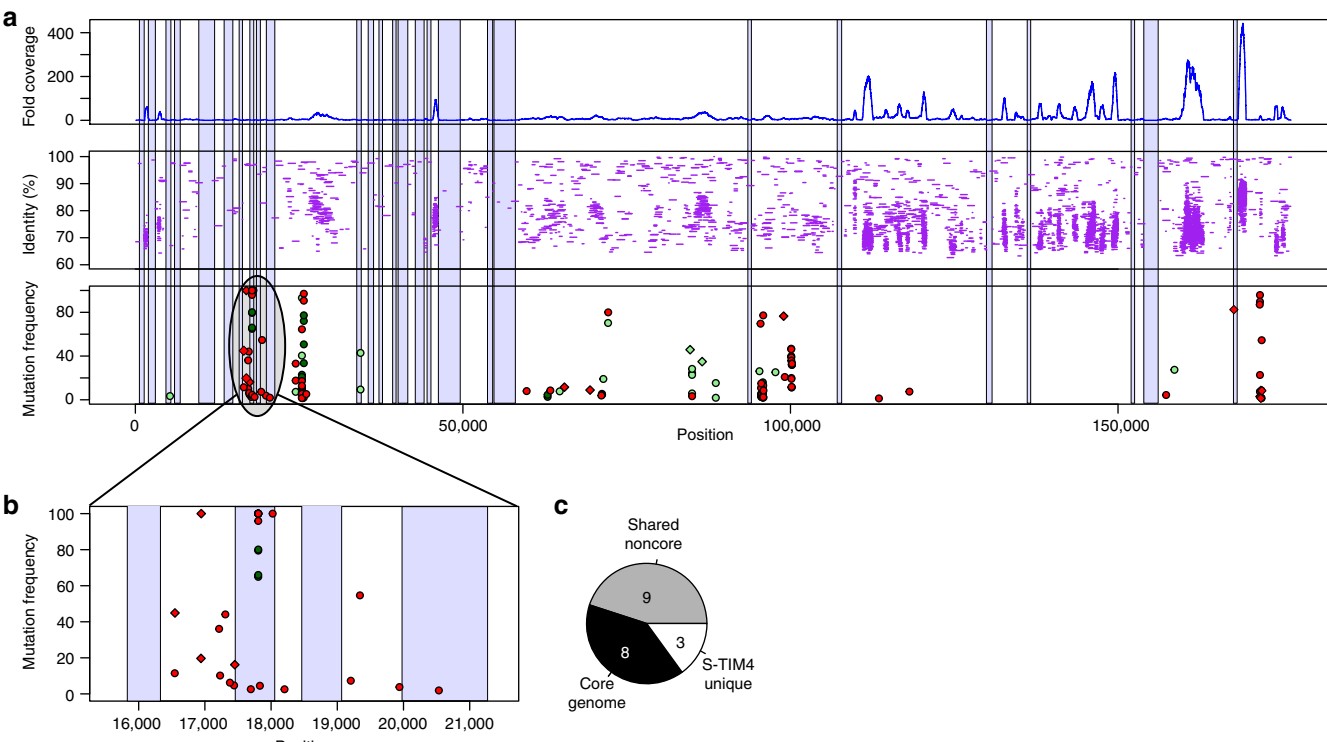

**Fig. 5** Variable genomic regions are not enriched in mutations. **a** Mapping of metagenomic reads to the S-TIM4 genome reveals mutations in variable and conserved regions with no enrichment in either. The position along the genome is on the X-axis. The lower panel shows mutation frequencies for phage populations along the genome. Colors denote the type of host the viral population evolved in: MIT9515 (dark green), MED4 (light green), and WH8102 (red). The middle panel shows the mapping of metagenomic reads to the S-TIM4 genome with the percent nucleotide identity of each read to the viral genome. The upper panel shows the fold-coverage of the metagenomic reads along the genome. Hypervariable genomic regions are shaded in purple (i.e. >500 bp long, coverage lower than 20% of mean). **b** Zoomed-in view of a mutation-rich genomic region (positions 15,500–21,500, open reading frames 23–27) of the S-TIM4 genome. **c** Adaptation of S-TIM4 to hosts occurs mostly by mutations in genes found in other cyanomyophages. Genes are classified as unique in S-TIM4 (have no close homologs in other phages), belonging to the cyanomyophage core genome (present in all studied cyanomyophage genomes), or shared noncore (appear in some cyanomyophage genomes). Raw data can be found in Supplementary Tables 2, 4

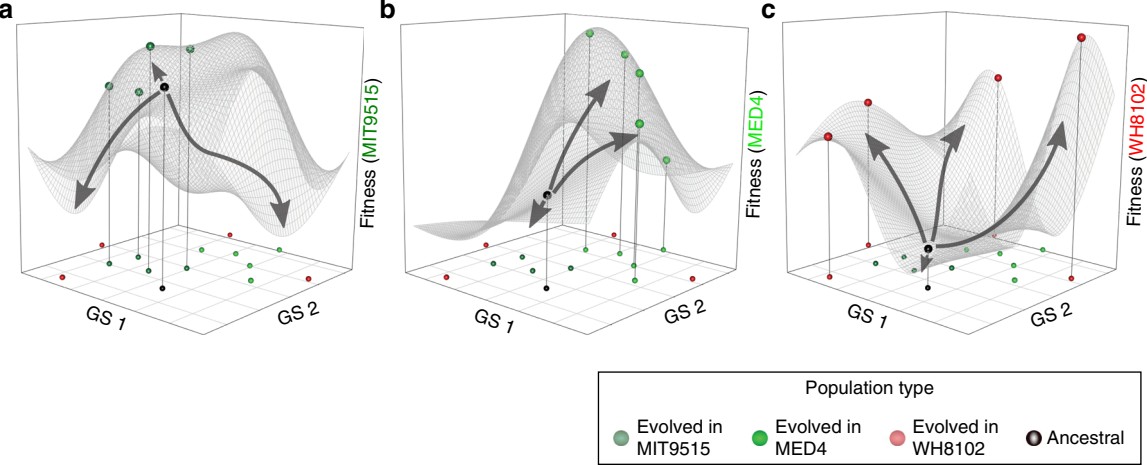

**Fig. 6** Model for the influence of different hosts on the evolution of viral populations. Each panel represents the infectivity landscape of the evolved and ancestral phage populations when infecting **a** the optimal host, *Prochlorococcus* MIT9515, **b** sub-optimal host *Prochlorococcus* MED4, and **c** sub-optimal host *Synechococcus* WH8102. In all infectivity landscapes the GS1-GS2 plane (GS, Genomic Space) is identical and corresponds to pair-wise genetic distances between populations. The Z-axis shows the fitness of phage populations, based on the ability to infect each host (see Methods). Predicted fitness surfaces (mesh over the genomic plane) were calculated based on the infectivity of all phage populations when infecting each of the cyanobacterial hosts. Spheres located on the GS1-GS2 plane show the location of each phage population in the genotypic space. Only the ancestral population and the populations that evolved in the host used to measure the infectivity appear as spheres on the fitness surface. When the optimal host is abundant, phage proliferation occurs mainly by infections of this host (panel **a**). The fitness of the ancestral population is high and most mutations result in lower fitness, and are therefore purified by negative selection (downwards arrows). Only a few mutations result in higher fitness and are positively selected (upwards arrow). Therefore, the evolved populations remain close to the ancestral population, both genotypically and phenotypically. Upon reduction in the abundance of this host, infections of "sub-optimal" hosts become frequent (panel **b**). When the fitness of the ancestral phages on this host is lower, compared to the fitness in the optimal, phage evolution on sub-optimal hosts results in a higher fraction of positively selected mutations, increased fitness of evolved populations and higher genotypic diversification. Often the only potential host is poorly infected by the ancestral phage (panel **c**). When the ancestral population evolves in this host, high numbers of mutations are positively selected, leading to rapid divergence into distinct phage populations, both phenotypically and genotypically. This demonstrates how the type of bacterial host impacts the diversity of phage populations. Raw data can be found in Supplementary Table 2 and Supplementary File 1

the availability of an "optimal" host is high, phage proliferation will occur mainly through infections of this host, as predicted by previous theoretical work[49]. As the phage is most adapted to such infections, the majority of the mutations that occur in the phage population result in lower infectivity (which likely directly reflects phage fitness), and their frequency is thus kept low as a result of purifying selection. Only a few mutations result in higher fitness when phages infect the optimal host strain, and the population converges closer to the maximal fitness point (Fig. 6a). When availability of the optimal host declines either due to selective sweeps that result from environmental change, the acquisition of resistance by the optimal host, or the killing off of the host due to phage infection, then phage proliferation occurs by infection of sub-optimal hosts and the fitness of the phage population is expected to decrease. As a result, the population adapts to sub-optimal hosts: less mutations are eliminated by purifying selection and distinct sets of mutations are positively selected (Fig. 6b, c). These mutations are mostly in genes responsible for host recognition, attachment, and infection. Support for this part of our model comes from a recent study of phage adaptation to a new host in the mouse gut when the preferred host bacterium is absent[16]. The positive selection of genotypes carrying these mutation sets results in rapid diversification and an increase in fitness on this host. This would be the initial step in the separation of the phage population into distinct subpopulations.

A number of studies suggest that extinction of an abundant microbial host and the proliferation of a rare host results in an increase in the abundance of rare viral types (reviewed in ref. [48]). Based on our findings, we suggest that changes in host availability not only change the abundance of different viruses[50], but are also

a key factor in the creation of the extensive degree of viral diversity observed in the environment.

## Methods

**Cyanobacterial and cyanophage strains and growth conditions.** *Synechococcus* sp. strain WH8102 (NCBI:txid84588), *Prochlorococcus* sp. strain MED4 (NCBI: txid59919), and *Prochlorococcus* sp. strain MIT9515 (NCBI:txid167542) were used in this study. *Prochlorococcus* strains were grown in PRO99 medium[51] on Mediterranean seawater base. *Synechococcus* strains were grown in ASW (artificial seawater) medium[52]. Cultures were grown at 22 °C under cool white light at an intensity of 10 µmol photons m$^{-2}$ s$^{-1}$, with a 14:10 h light:dark cycle.

The selection of bacterial strains used in this study was based on the host range of S-TIM4[6] (NCBI accession MH512890) and Syn19[12] (NCBI Reference Sequence: NC_015286.1) phages, and was confirmed by infection assays of potential hosts.

**Obtaining an isogenic phage and phage growth.** S-TIM4 plaques were formed on lawns of *Synechococcus* WH8102 by pour-plating mixtures of ~10$^8$ cells with viral lysates on ultra-pure low-melting-point agarose (Invitrogen) at a final concentration of 0.28% in ASW, in a similar manner to that described previously[53]. Similarly, Syn19 plaques were obtained by infecting *Synechococcus* sp. strain WH8109 cells. Plaques were isolated and propagated in liquid cultures of exponentially growing WH8102 cells (S-TIM4) and WH8109 (Syn19). PCR was performed on propagated clones, aiming to amplify the T4-like *g20* gene. *g20* PCR products were excised from 2% agarose gels and purified with a MinElute gel extraction kit (Qiagen) before Sanger sequencing was carried out at Hylabs (Rehovot, Israel). One S-TIM4 clone and one Syn19 clone were further propagated to yield the ancestral S-TIM4 and Syn19 strains, after verifying their classification using the sequenced *g20* gene.

**Evolutionary experiment.** Initially, aliquots of the ancestral populations were used to infect 1 ml liquid cultures of exponentially growing host cells in 48-well microplates. S-TIM4 was used to infect five bacterial cultures of each of the hosts: *Synechococcus* sp. WH8102, *Prochlorococcus* sp. MED4, and *Prochlorococcus* sp. MIT9515. Syn19 was used to infect five bacterial cultures of the *Synechococcus* WH8102 host. Cyanobacterial cultures were monitored by measuring chlorophyll *a*

fluorescence using a Synergy2 Microplate Reader (Ex/Em: 440/680 nm; BioTek) which is a proxy for cell density[51]. Once the reduction in the cell densities of infected cultures ceased (while growth of the uninfected control continued), viral lysates were filtered (0.22 μm Millex GV syringe filter, Millipore (Cork, Ireland)). This enabled us to avoid the possibility of transferring resistant cells, and as a result, allowing coevolutionary dynamics. Filtered lysates were stored in glass bottles at 4 °C. To start a new lysis cycle, a fixed volume of 50 μl of the filtered lysate was used to infect 1 ml of the naïve cyanobacterial hosts of the same strain each lysate grew on. Overall, 15 cycles were conducted, the length of individual infection cycles ranged between ~1 and 3.5 weeks.

**DNA extraction and sequencing.** Ancestral and evolved viral lysates were propagated in liquid cultures of the host they evolved on to a final volume of 5–25 ml. Cell debris was removed by centrifugation (10,000 × g at 20 °C for 15 min) and filtered using a 0.22 μm syringe filter, as described above. Filtered lysates were concentrated by centrifugation at 4 °C (Amicon Ultracel 30 KDa, Millipore). Possible cellular DNA contaminations were digested using either DNase I (Sigma) or Turbo DNase free (Thermo Fischer Scientific), RNA was digested using RNase A (Sigma). Phage DNA was extracted and purified using a phenol-chloroform method as described previously[54].

Phage DNA was sheared using the E220 ultrasonicator (Covaris, Woburn, MA, USA). For each evolved population, two DNA sequencing libraries were constructed using NEBNext Ultra DNA Library Prep Kit (E7370L) and NEBNext Multiplex Oligos (E7600S) (New-England Biolabs, MA, USA). For ancestral populations one library per population was constructed. Paired-end (2 × 100) DNA libraries were sequenced at the Technion Genome Center, using an Illumina HiSeq 2500 sequencer (Illumina, San-Diego, CA, USA). In total, we managed to successfully sequence all populations, except for one S-TIM4 population evolved on MIT9515 population and one Syn19 population.

**Genomic data analysis.** Quality of DNA reads was assessed using the FastQC software[55]. Adapter sequences were removed using Cutadapt[56] with a minimum read length of 51 base-pairs. All sequencing libraries were mapped to the corresponding reference genome to detect mutations using the BreSeq software[57]. Ancestral populations were analyzed in clonal mode and identified mutations were used to update the reference genomes. Evolved populations were analyzed in population mode (BreSeq -p flag), to identify mutations present in fractions of the viral populations. To avoid strand bias (i.e., erroneous identification of polymorphism identified only on one of the strands), minimal coverage of ten reads on each of the strands was required for each polymorphism variant to be included in the software output. To eliminate false positives as a result of library preparation and sequencing errors, only mutations with frequency ≥1% in each of the two libraries originating from each evolved population were used. The average of the frequencies from both of the libraries was used in downstream analyses. As one of the libraries for population MIT9515 (#2) was poorly sequenced, we only considered one library for this population. We used the SPAdes genome assembler[58] to detect cellular DNA contaminations and long insertions and deletions, with the –cov-cutoff 10 flag. We discovered that one of the S-TIM4 populations, evolved in WH8102, was highly contaminated with bacterial DNA of an undetermined source and was therefore excluded from downstream analysis.

Overall, 14 S-TIM4 populations were successfully sequenced and included in our analyses: one ancestral population, four populations evolved in *Prochlorococcus* MIT9515, four populations evolved in *Synechococcus* WH8102 and five populations evolved in *Prochlorococcus* MED4 host. Five Syn19 populations were included in the analysis, one ancestral population, and four populations evolved in WH8102.

**Infectivity assays of ancestral and evolved populations.** To start all infectivity tests at the same point in time, aiming at minimal decay of the viruses, a 100 μl of each lysate was added on the same day to 5 ml of the host on which it had evolved (at mid-log stage). Upon completion of lysis, each lysate was filtered through a 0.2 μm pore-sized Acrodisc Syringe Filter (Pall Corporation) into a glass container and stored at 4 °C. Shortly after, each lysate was quantified by qPCR, relative to a TOPO-PCRII linearized plasmid containing a specific 300 bp insert of the *g20* portal gene of the published viral sequence.

Tests for determining the lysis rate of each viral population started on the same day for all populations, using the same number of viruses and concentrations of cyanobacterial cultures, at mid-log stage and at two VpC (Viral particles/host cell) ratios of 0.1 and 3. Enumeration of cyanobacterial cultures was conducted using the HTS (High Throughput Sampler) option of an LSRII flow cytometer (Becton Dickenson), using three technical repeats. Lysis tests were carried out in 96-well microtiter plates, with each three wells representing three biological replicates per viral population. For controls 1 μl of growth medium was added in lieu of lysate. The final volume per well was 200 μl. Cultures were monitored daily by measuring of chlorophyll *a* fluorescence using a Synergy2 Microplate Reader (Ex/Em: 440/680 nm; BioTek).

**Phenotypic profiles and phenotypic distance tree.** To create phenotypic profiles for the evolved and ancestral populations (Fig. 3c), the infectivity of each S-TIM4

population was determined when infecting each cyanobacterial host at ratios of 0.1 and 3 viral particles/host cell, as detailed in Eq. (1).

$$\text{Infectivity}_{i,j,\text{VpC}} = \frac{\langle \text{CD}_{i,j,\text{VpC}} \rangle}{\langle \text{CD}_{c,j} \rangle}, \quad (1)$$

where *i* is population *i*; *j* is host *j*; VpC is viral particles/host cell; CD is cell density, 10 days post-infection; *c* is control (uninfected) cultures of host *j*.

Each population is represented by a vector of six elements, consisting of the infectivity values of the population on all three hosts, at both VpC ratios. Pair-wise phenotypic distances were calculated between all S-TIM4 populations, by implementing the Canberra distance[59], using the amap R-package[60]. The resulting distance matrix was used as input for the PHYLIP NEIGHBOR software[61] to calculate a neighbor-joining tree. Tree visualization was conducted using the iTOL web server[62].

**Genetic distance calculation and genotypic distance tree.** To calculate pair-wise genotypic distances between S-TIM4 populations, each population was represented by a vector of 86 elements, consisting of the mutation frequency in each of the 86 mutated genomic locations identified in previous steps (e.g., the ancestral population is represented by a vector containing 86 elements of value 0, indicating the absence of mutations for this population). Pair-wise distance was calculated as the Euclidean Distance between each two vectors (Eq. (2)) as it allows for calculations of distances between elements with zero value.

$$\text{ED} = \sqrt{\sum_{i=1}^{86} (Fx_i - Fy_i)^2}, \quad (2)$$

where $Fx_i$ is frequency of mutation *i* in population *x*; $Fy_i$ is frequency of mutation *i* in population *y*.

The computed Euclidean distances were used as input for the PHYLIP NEIGHBOR software[61] to calculate a neighbor-joining tree, where the ancestral populations were defined as outgroup. Tree visualization was conducted using the iTOL web server[62].

**Classification of genes as early, middle, or late genes.** Prediction of gene expression phase was based on that found by Doron et al.[31]. for Syn9. S-TIM4 genes were compared to Syn9 genes using NCBI BLAST+ program suit[63] and S-TIM4 genes were classified as early, middle or late based on that of the closest homolog of each gene in the genome of Syn9.

**Mass spectrometry identification of virion proteins.** Identification of virion proteins was performed using liquid chromatography/mass spectrometry (LC-MS/MS), as described previously[64]. Briefly, ancestral S-TIM4 particles were CsCl purified and digested by modified trypsin (Promega). Digested and purified peptides were analyzed by LC-MS/MS, using an ion-trap mass spectrometer (Orbitrap, ThermoScientific). Data were analyzed using Sequest 3.31 software searching against the S-TIM4 genome and *Prochlorococcus* sp. MIT9515 genome, the latter to eliminate false identification of host peptides.

**Recruitment of marine metagenome to S-TIM4 genome.** To detect hypervariable genomic regions, we recruited the GOS metagenome[44] to the S-TIM4 genome, using the approach described by Rusch et al.[44]. We used the GOS dataset as it contains high-quality Sanger sequences with average length of ~1000 bp. Briefly, we mapped reads aligned to the S-TIM4 genome for more than 300 bp at 65% identity, with less than 25 unaligned bases allowed on either end. We also used reads aligned over less than 300 bp but with over 100 bp at >65% identity, with less than 20 unaligned bases allowed on either end. Some reads were successfully mapped but their mate-pairs were not mapped under the specified conditions. In these instances, if the mate sequence was successfully aligned for >80% of its length the two reads were recovered and recruited to the S-TIM4 genome.

We calculated the position-specific fold-coverage of metagenomic reads along the S-TIM4 genome using the R-package IRanges[65]. Hypervariable regions were defined as regions with fold-coverage <20% of the median coverage, for ≥500 base-pairs, as was defined previously[35].

**Fitness landscape model.** Dimensional reduction of the genomic profiles (t-distributed stochastic neighbor embedding (t-SNE)[66]) was conducted using R-package tsne[67] with the genomic Euclidean distance matrix as input. The two-dimensional reduction of the genotypic space was used to generate the three identical X~Y planes shown in Fig. 6. Fitness values (Z-axis) are based on the infectivity of the viral populations when interacting with each of the three cyanobacterial hosts. To create the fitness landscape we represented each

population as the 1/infectivity of each host at VpC = 3 (Eq. (3)).

$$Z_{i,j,(\text{VpC}=3)} = \frac{\langle \text{CD}_{c,j} \rangle}{\langle \text{CD}_{i,j,(\text{VpC}=3)} \rangle}. \tag{3}$$

Further, we applied the inverse distance weighted interpolation (kriging) approach[68] to compute Z-axis values for each coordinate on the X~Y grid using the R-package gstat[69] and smoothed the surface using the generalized additive model (gam) method[70]. Further, we plotted the locations of all S-TIM4 populations on the X~Y plane. Additionally, for the ancestral population and the populations that evolved on the corresponding host, we also plotted the locations on the fitness landscape surface.

**Code availability**. Custom code used in this work is available at: https://github.com/henav/cyanomyo_evo.

## Data availability

All sequencing libraries, of both the ancestral and evolved phage populations, are deposited in NCBI SRA database (NCBI BioProject accession: PRJNA478496). The data supporting the findings of this study are available within the Article and Supplementary files, or available from the authors upon request.

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

## Acknowledgements

We would like to thank Sarit Avrani for providing cyanophage S-TIM4 and extensive help, Daniel Schwartz for providing raw data for Supplementary Fig. 1, and Omer Nadel for preparing phage isolates for mass-spectrometry analysis. This work was supported by the Louis and Lyra Richmond Memorial Chair in Life Sciences (to O.B.) and by a grant from the Simons Foundation (SCOPE Award 329108 to D.L.).

## Author contributions

H.E., Y.M.-G., D.L and O.B. designed the project. H.E. and S.K. performed laboratory experiments. H.E. performed bioinformatic analysis and wrote the manuscript with significant contributions from all authors.

## Additional information

**Competing interests:** The authors declare no competing interests.

