## [Peer Review file · Nature Communications]

Reviewers' comments:

Reviewer #1 (Remarks to the Author):

Review

Adaptation to sub-optimal hosts is a driver of viral diversification in the ocean by Hagay Enav et al.

Reviewer: Hiroyuki Ogata (Professor/Kyoto University)

The authors performed an experimental evolution study of cyanophages using three host strains with differences in efficiency of infection and showed that infection of sub-optimal hosts acts as a driver for rapid diversification of phage populations. More specifically, the authors used S-TIM4 (a generalist cyanomyovirus) which shows different efficiency levels against (i) Prochlorococcus MIT9515 (optimal efficiency), (ii) Prochlorococcus MED4 (middle efficiency) and (iii) Synechococcus WH8102 (least efficient). Fifteen rounds of serial infection were carried out against these three host strains to generate evolved S-TIM4 populations. The authors showed that viruses forced to infect the least adapted hosts ("sub-optimal hosts") exhibited populations with the largest number of mutations in their genomes (i.e. rapid diversification) along with the enhanced phenotypic alterations (an increased efficiency of infecting WH8102 with a reduced efficiency of infection to other host strains). Interestingly, most of the mutations were found in structural genes with many non-synonymous SNPs (89% of SNPs). Furthermore, the authors show that most of the non-synonymous SNPs improve the adaptation to the anticodons of host tRNAs. Detailed analysis of the locations of mutations suggest that they are not confined in either conserved or variable regions of cyanophage genomes.

I find this study interesting because this clearly demonstrates that shifts in host composition in a natural community, which may lead to the increase of "sub-optimal hosts", potentially accelerate adaptive evolution of existing viral population. This could have been speculated based on previous observations but has been missing a clear experimental validation. Also, the fact that most of the mutations fell in structural genes (Table S3) is impressive and underscores the importance of adaptive evolution (and not random genetic drift) in the survival of viruses encountering non-optimal environment (hosts).

I have a few minor comments on the submitted manuscript that the authors may want to address to improve the manuscript.

Line 99, 107: Of SNPs, 89% were non-synonymous. This is larger than 75% expected from total lack of positive/negative selection. The difference was statistically significant ($p=3e-3$). I agree that some of these non-synonymous mutations were probably positively selected. However, the difference (75% vs 89%) does not prove that all observed non-synonymous mutations resulted from positive selection. In the same sense, the label in Fig. 2a "Positive selection 98%" is a too much simplified expression, although in the legend this is described better. It is also known that slightly deleterious non-synonymous mutations can accumulate rapidly during a short period of evolutionary time but may be swept later (ex. Evolutionary biology: relativity for molecular clocks. Penny D. Nature. 2005 Jul 14; 436(7048); Rocha EP, Smith JM, Hurst LD, Holden MT, Cooper JE, Smith NH, Feil EJ. Comparisons of dN/dS are time dependent for closely related bacterial genomes, J Theor Biol, 2006, vol. 239 (pg. 226-235)). The positive selection may be more strongly suggested by the restricted locations of mutations (Fig. 3d; Table S3) and may have to be emphasized at the stage when these Fig. 3d and Table S3 are mentioned.

Line 467: remove one "evolved in".

Line 40: I think it is informative to cite recent studies revealing the role of cyanophages in the control

of the diel cycle of cyanobacteria in natural environments such as Yoshida et al. *Locality and diel cycling of viral production revealed by a 24 h time course cross-omics analysis in a coastal region of Japan*. ISME J., doi:10.1038/s41396-018-0052-x (2018).

I am not sure if the reference genome sequence data of S-TIM4 is available (I could not find it in GenBank). If its genome is not deposited to a public db please deposit. Also same for other genomic data including the newly generated S-TIM4 and Syn19 population data, if not yet done.

Line 87: I do not understand the following part of sentence: "using the average mutation frequencies from two sequencing libraries per evolved population" Could you make this clearer?

Does S-TIM4 encode tRNA genes? Could you mention about the relationships between observed mutations and anticodons of viral tRNAs?

Line 275: "only 14.5%", why only? What is the expected proportion in under the null hypothesis?

Line 472 "phage" -> "Phage"

Is the "wild type" S-TIM4 can be really called as a wild type?

Reviewer #2 (Remarks to the Author):

The manuscript by Enav et al explores how marine cyanophages with broad host ranges might genetically adapt to better infect sub-optimal hosts. Using an experimental evolution approach, the authors found sets of mutations in cyanophages that influenced their ability to infect particular hosts. They then show that these genetic changes in viral populations correspond with an increased rate of viral lysis on "sub-optimal" hosts. Interestingly, in some cases, adaptations to better infect one host limited the ability of the cyanophage population to infect its other hosts. Overall, this study provides relevant and interesting results that could help explain the genetic diversity observed in natural populations of cyanophages. Cyanophages can be a major source of mortality for marine cyanobacteria, thus knowledge about how viral host ranges can change (particularly for viruses such as these that can infect hosts of different genera) is important in understanding marine food webs.

Specific questions, comments and concerns:

1. I really like the experimental evolution approach used in this study. Results from these types of lab experiments could help researchers interpret the genetic diversity observed in natural cyanophage populations. Recently, there have been quite a few studies that examined genomic diversity among cyanomyophage isolates from natural communities and populations (for example, Deng et al *Nature* 2014, Gregory et al *BMC Genomics* 2016, Cordero EM 2017, Crummett et al *Virology* 2016, Marston & Martiny EM 2016). The results reported by Enav et al should be put into context of what is already known about genomic diversity in cyanomyophage isolates from natural populations. The Gregory et al study is especially relevant because it examined the host ranges of natural cyanomyophage isolates belonging to the same population. Almost all of these articles include hypotheses about how cyanomyophages might adapt to hosts, which is exactly what this study is testing.

2. Line 68: In this study, "optimality" is being defined as the "efficiency" in which the phages infect their hosts. In the methods section, it appears that phage "infection efficiency" is based solely on the rate/amount of host lysis. If this study is trying to understand how viral populations are adapting to

suboptimal hosts, then viral fitness, rather than host lysis, should be measured (where viral fitness = 'production of phage progeny over multiple rounds of infection' as in Schwartz and Lindell ISMJ 2017). I am concerned that the host lysis rate is not a good proxy for viral fitness. qPCR was done to titer viral lysates prior to the infectivity assays, it could also be used to titer lysates during and at the end of the assay. If "host lysis rate" is what is going to be used to as a proxy for viral fitness, it should be made clear at the beginning of the manuscript (lines 68, 88, etc.) and data illustrating how host lysis relates to viral fitness should be included.

3. Line 77: Please clarify what you mean by "thousands of viral genome replication events". Is this referring to generation times?

4. Line 80 and Methods: Additional information about the evolutionary experiment needs to be included. How much viral lysate was used to infect the liquid cultures in each round of evolution? Were the aliquots titered in each round or was just a set volume (e.g., 10 ul) transferred? Approximately how long did it take for "complete lysis" of the cultures – how was "complete lysis" determined? How many viral generations occurred between transfers?

5. Line 91: The GenBank accession numbers of the S-TIM4 and Syn19 genome sequences need to be included. If the S-TIM4 genome was sequenced as part of this study, then the methods for assembly and annotation should be included. In addition, all of the sequence reads from each of the evolved populations should be deposited in a public database and the information on how to access those data needs to be included in the manuscript.

6. Genomic Data Analysis: Please clarify "Minimal coverage of 10 reads/strand was required to include polymorphisms in software output". Is this 10 reads/strands in all, or 10 reads/strand of the variant polymorphism? What was the average coverage of the genome for each of the sequenced populations? Even by sequencing two libraries, a >1% threshold for polymorphisms seems low – particularly if the overall coverage is low. Looking at the list of polymorphisms, there are quite a few below 10%, especially for populations evolved on WH8102. Even if these SNPs are not sequencing errors, how important are SNPs at very low frequency likely to be in influencing overall infection efficiency of the population? More weight should be put on polymorphisms that have greatly increased in frequency in the population and thus are more likely to be under selection. These polymorphisms will also contribute more to the increased infectivity of the viral populations. (for example, a recent article analyzing E coli LTEE populations used a 10% threshold for mutations: Good et al Nature 2017).

7. Line 100, Figure 4: It would be helpful to standardize how genes are being referred to in the manuscript. In some cases, the ORF number is used, in some cases a gp# is used and in others, the name of the protein. The relationship among these is not clear unless you're looking at TableS3. In Fig.4 – all three designations are used, which is confusing. I would always include the ORF number and then the gp# or protein name if it provides some important additional information.

8. Line 120 and Methods: In the methods section, it states that in addition to S-TIM4, Syn19 was also evolved on 5 cultures of each of two hosts (WH8109 and WH8102). However, the only mention of these results is in line 120: "These populations accumulated 4-6 mutations per population". No data are shown. The inclusion of the results for the evolution of Syn19 in these two hosts would strengthen the paper and provide an interesting comparison for the results obtained with S-TIM4. It would be useful to know if the results obtained for Syn19 show similar trends to S-TIM4. If the Syn19 results are not going to be discussed, then the method section should be revised.

9: Line 228: The result of 15 synonymous mutations in one 120 bp segment that optimizes codon usage is very interesting. It is more striking that all of these mutations occur together on the same

sequencing reads (Fig S1). It seems very unlikely that through the process of random mutation during 15 rounds of evolution, 15 synonymous mutations in one 120 bp segment would occur all in the same isolates. If these reads are not contamination, perhaps there has been recombination of this segment with another repeated segment elsewhere in the genome? The authors should propose/explain how these mutations likely arose and rule out possible contamination.

10: Line 235: Does S-TIM4 carry any tRNAs? If so, which ones?

11. Line 264: This statement should be qualified –for example, in “all cyanomyophages” analyzed in ref. 37. As more cyanomyophages are sequenced, what is considered the “core” genome will change.

12. Lines 270-280: As stated in comment 1, the results of this study could be very informative in trying to understand genomic diversity in natural viral populations. So again, the results should be put in context and compared to what is already known about cyanomyoviral genomic diversity in natural populations. In this experimental setup, the mutations contributing to host adaptation are “not preferentially mediated through mutations in hypervariable regions”. In natural communities, however, cyanophages probably adapt to new hosts not just by mutations, but also through homologous or non-homologous recombination with other related phages or even hosts. In this study, there was only one phage genotype present, therefore allelic recombination, or the acquisition of new genes, was not possible. A sentence should be added to this section acknowledging the limitations of the presented results. In addition, could the proposed model also incorporate, for example, the acquisition of a new AMG that might increase adaptation to a particular host - to give the model a more holistic picture of phage adaptation?

Reviewer #3 (Remarks to the Author):

The paper is an interesting contribution by a well known research group that seeks to examine how diversification in viral genomes occurs through the infection of hosts with different efficiencies. In general I like the idea of the study but I have a few questions.

The authors use the term "sub-optimal" but never really define this in a quantitative sense. Can they do this? It would seem to me that a qualitative definition (which is how I read the paper) is not as strong.

Page three. "In natural habitats, the abundance of cyanobacterial hosts...". These are a mix of hosts and non-hosts, right? Or putative hosts as in some cases the physiological conditions / environmental conditions need to be right to make things work.

Line 54 and elsewhere. The authors use "in order to" a few times in the paper and they can delete almost every one of these.

Line 55. Would forming a lysogen not also be a solution to this problem?

Line 68. Optimality refers to efficiency. You need to define efficiency here. Is this rate of contact success? Rate of infection of individuals? Fitness (particles produced per infection, or infectious particles produced per infection). I am trying to illustrate that different readers could take this differently and you need to bring them along with you on this.

Line 89, I am left wondering about "wild-type"...presumably this has been in someone's lab for many generations? Are there effects from previous rounds of culture that could influence your observations (same with the hosts for that matter).

Line 94. I do not see how supplemental table 1 shows efficiency. I see fluorescence readings. Is efficiency defined as the amount of a population killed (if so it would be nice to back this up with cell counts)

Line 107. Is it possible the hosts just have better proof-reading or repair mechanisms? How would that play into your observations. I realize you mention this a little later on this page but the mention is very brief and I am not sure how comparable these two viruses are. Just playing devil's advocate here.

Line 149. I am not sure what "clustered most differently" means....please clarify

Line 211. The authors keep mentioning positive selection. But I am not sure what negative selection would look like...I think that needs to be clarified.

Line 242. Is this DUF region also found in hosts?

Line 256. Confusing heading. Occurs in conserved and variable regions....so everywhere? Random?

Line 287. again highest efficiency? So lysis? Fewer resistant cells?

Line 309. Matteson et al (FEMS MICROB ECOL 2013) show that when *Synechococcus* massively decreases in the water column the viruses massively decrease as well in the days that follow...This supports the first part of your final statement (alas, they do not appear to have done the sequencing that would really prove your point).

Answers to Reviewer #1 comments:

The authors performed an experimental evolution study of cyanophages using three host strains with differences in efficiency of infection and showed that infection of sub-optimal hosts acts as a driver for rapid diversification of phage populations. More specifically, the authors used S-TIM4 (a generalist cyanomyovirus) which shows different efficiency levels against (i) Prochlorococcus MIT9515 (optimal efficiency), (ii) Prochlorococcus MED4 (middle efficiency) and (iii) Synechococcus WH8102 (least efficient). Fifteen rounds of serial infection were carried out against these three host strains to generate evolved S-TIM4 populations. The authors showed that viruses forced to infect the least adapted hosts (“sub-optimal hosts”) exhibited populations with the largest number of mutations in their genomes (i.e. rapid diversification) along with the enhanced phenotypic alterations (an increased efficiency of infecting WH8102 with a reduced efficiency of infection to other host strains). Interestingly, most of the mutations were found in structural genes with many non-synonymous SNPs (89% of SNPs). Furthermore, the authors show that most of the non-synonymous SNPs improve the adaptation to the anticodons of host tRNAs. Detailed analysis of the locations of mutations suggest that they are not confined in either conserved or variable regions of cyanophage genomes.

I find this study interesting because this clearly demonstrates that shifts in host composition in a natural community, which may lead to the increase of “sub-optimal hosts”, potentially accelerate adaptive evolution of existing viral population. This could have been speculated based on previous observations but has been missing a clear experimental validation. Also, the fact that most of the mutations fell in structural genes (Table S3) is impressive and underscores the importance of adaptive evolution (and not random genetic drift) in the survival of viruses encountering non-optimal environment (hosts).

I have a few minor comments on the submitted manuscript that the authors may want to address to improve the manuscript.

Line 99, 107: Of SNPs, 89% were non-synonymous. This is larger than 75% expected from total lack of positive/negative selection. The difference was statistically significant ($p=3e-3$). I agree that some of these non-synonymous mutations were probably positively selected. However, the difference (75% vs 89%) does not prove that all observed non-synonymous mutations resulted from positive selection. In the same sense, the label in Fig. 2a “Positive selection 98%” is a too much simplified expression, although in the legend this is described better. It is also known that slightly deleterious non-synonymous mutations can accumulate rapidly during a short period of evolutionary time but may be swept later (ex. Evolutionary biology: relativity for molecular clocks. Penny D. Nature. 2005 Jul 14;436(7048); Rocha EP, Smith JM, Hurst LD, Holden MT, Cooper JE, Smith NH, Feil EJ. Comparisons of dN/dS are time

dependent for closely related bacterial genomes, J Theor Biol, 2006, vol. 239 (pg. 226-235)). The positive selection may be more strongly suggested by the restricted locations of mutations (Fig. 3d; Table S3) and may have to be emphasized at the stage when these Fig. 3d and Table S3 are mentioned.

>> We agree that the high fraction of non-synonymous SNP's does not prove that all were positively selected and realized that our initial statement was misunderstood. When discussing the observed enrichment of non-synonymous SNP's in the original manuscript we aimed to emphasize the role played by positive selection, in the adaptation to suboptimal hosts. Given that this was unclear we have done the following:

* Following the reviewer's suggestion we have changed figure 2a, removing the inner circle, and the sentence discussing positive selection from the legend.

* In lines 113-114 we have added a sentence discussing our findings in light of the model suggested by Rocha et-al and cited this manuscript. It reads: "...when considering that some of these mutations could result from genetic hitch-hiking and possibly be swept later".

* We have changed the title of the paragraph in line 103 to: "Positive selection plays a major role in the emergence of mutants in cyanophage populations".

* We have rephrased the paragraph discussing the restricted location of mutations to emphasize the role played by positive selection (line 205-207). It reads: "five of the same six mutations that were identified in the phage populations evolved in *Prochlorococcus* MIT9515 were also found in nearly all of the other phage populations (Figure 3d, Table S2), further emphasizing the role played by positive selection."

Line 467: remove one "evolved in".

>> Thank you for noticing this. We have corrected this typo.

Line 40: I think it is informative to cite recent studies revealing the role of cyanophages in the control of the diel cycle of cyanobacteria in natural environments such as Yoshida et al. Locality and diel cycling of viral production revealed by a 24 h time course cross-omics analysis in a coastal region of Japan. ISME J., doi:10.1038/s41396-018-0052-x (2018).

>> We thank the reviewer for this suggestion. Accordingly, we have added to the introduction the findings by Yoshida et al. as well as the relevant citation (line 43-44).

I am not sure if the reference genome sequence data of S-TIM4 is available (I could not find it in GenBank). If its genome is not deposited to a public db please deposit. Also same for other genomic data including the newly

generated S-TIM4 and Syn19 population data, if not yet done.

>> We have deposited the genome sequence of the S-TIM4 phage, and added the accession numbers of S-TIM4/Syn19 genomes to the methods section. Additionally, we have submitted all sequencing data to NCBI-SRA database, and added the accession numbers in the methods section

Line 87: I do not understand the following part of sentence: “using the average mutation frequencies from two sequencing libraries per evolved population” Could you make this clearer?

>> We have changed this sentence to be more understandable. It now reads (lines 90-92): “we performed whole population genome sequencing, with two sequencing libraries per evolved population and used the average frequency of mutations identified in both technical repeats.”

Does S-TIM4 encode tRNA genes? Could you mention about the relationships between observed mutations and anticodons of viral tRNAs?

>> Thank you for raising this point. Indeed, S-TIM4 encodes 3 tRNA genes. We agree that this is very important to note and added this information including a description of the relationship between the tRNA genes and the observed synonymous mutations, in the appropriate section (lines 239-242). It reads: “While none of these mutations results in adaptation to any of the three tRNA genes carried in the S-TIM4 genome, most mutations (12 out of 15) result in optimizing the codons to the host tRNA genes, thus potentially increasing the translation rate and accuracy of the ORF108.”

Line 275: “only 14.5%”, why only? What is the expected proportion in under the null hypothesis?

>> Under the null hypothesis, we expected enrichment of mutations in the hypervariable regions. We realize that this was not clear and we now elaborate on this issue stating that these regions compose 16.6% of the genome, to better demonstrate that no such enrichment was observed. Sentence added (lines 294-295): “Interestingly, only 14.5% of the S-TIM4 mutations are in hypervariable regions, while 16.6% of the genome is defined as hypervariable”.

Line 472 “phage” -> “Phage”

>> Thank you for noticing this mistake, the typo was corrected.

Is the “wild type” S-TIM4 can be really called as a wild type?

>> The S-TIM4 lysate that was used for this study is a relatively new cultured virus and did not go through many rounds of culturing in the lab. Nevertheless, we agree with this remark, and now refer to the “wild-type” population as the “ancestral” population.

Answers to Reviewer #2 comments:

1. I really like the experimental evolution approach used in this study. Results from these types of lab experiments could help researchers interpret the genetic diversity observed in natural cyanophage populations. Recently, there have been quite a few studies that examined genomic diversity among cyanomyophage isolates from natural communities and populations (for example, Deng et al. Nature 2014, Gregory et al. BMC Genomics 2016, Cordero EM 2017, Crummett et al. Virology 2016, Marston & Martiny EM 2016). The results reported by Enav et al. should be put into context of what is already known about genomic diversity in cyanomyophage isolates from natural populations. The Gregory et al. study is especially relevant because it examined the host ranges of natural cyanomyophage isolates belonging to the same population. Almost all of these articles include hypotheses about how cyanomyophages might adapt to hosts, which is exactly what this study is testing.

>> We agree that these studies are relevant, and that our results should be put into context of what is known about genomic diversity in cyanophage populations. We discuss our results in respect to the findings described in three of these publications (Marston and Martiny, 2016, Cordero, 2016, and Gregory et al., 2016) in lines 304-310. It now reads: "In a previous study, it was demonstrated that cyanophage ecotypes remain present in a natural habitat⁴⁵, possibly by rapid recombination events which lead to a stable persistence of the phage population and unstable association with the host ecotypes⁴⁷. It was also suggested that discrete population boundaries are initiated by sympatric niche differentiation and maintained by recombination⁴⁶. Based on our results, we support the latter, however we propose that mutation and genetic drift play a key role in the initial formation of distinct phage populations".

2. Line 68: In this study, "optimality" is being defined as the "efficiency" in which the phages infect their hosts. In the methods section, it appears that phage "infection efficiency" is based solely on the rate/amount of host lysis. If this study is trying to understand how viral populations are adapting to suboptimal hosts, then viral fitness, rather than host lysis, should be measured (where viral fitness = 'production of phage progeny over multiple rounds of infection' as in Schwartz and Lindell ISMJ 2017). I am concerned that the host lysis rate is not a good proxy for viral fitness. qPCR was done to titer viral lysates prior to the infectivity assays, it could also be used to titer lysates during and at the end of the assay. If "host lysis rate" is what is going to be used to as a proxy for viral fitness, it should be made clear at the beginning of the manuscript (lines 68, 88, etc.) and data illustrating how host lysis relates to viral fitness should be included.

>> Indeed, we are defining “optimality” as the “efficiency with which phages infect their hosts” and this is based on host lysis rate. While we understand the reviewer’s concern, a previous study has shown that host decline is an extremely good proxy for virus fitness, with the two being significantly correlated (Turner et al. 2012). Furthermore, Turner et al. (2012) point out that using such an assay enables a more high-throughput means, important for evolutionary studies such as ours, and provides additional information as to the effect on the host population. Therefore, while we agree with the reviewer that production of phage progeny over multiple rounds of infection would be a more direct means for addressing viral fitness, the measure we have used is appropriate. We find it to be particularly so for our study, where we address how evolving virus populations adapt to their hosts, as this assay allows visualization of the effect of the infection process on the hosts by the ancestral and evolved viruses.

Following the reviewer’s comment, we now cite this paper in our revised manuscript (line 69-73) pointing out the strong association between the decline of the host population and fitness of the phage populations. It now reads: “In this context, optimality relates to the efficiency with which the phages infect their cyanobacterial hosts, measured by the rate of decline of the host population. This has been shown previously to be directly correlated with phage fitness and encompasses multiple ways in which fitness can change, including rate of adsorption, time to lysis (latent period) and burst size²⁶”.

3. Line 77: Please clarify what you mean by “thousands of viral genome replication events”. Is this referring to generation times?

>> Thank you for pointing out the problem with this statement. We had indeed used this phrase to refer to the cumulative number of genomes replicated. However, after giving it more thought, we realize that since this is only a rough estimation it may be more confusing rather than informative. We have therefore removed this sentence from the text.

4. Line 80 and Methods: Additional information about the evolutionary experiment needs to be included. How much viral lysate was used to infect the liquid cultures in each round of evolution? Were the aliquots titered in each round or was just a set volume (e.g., 10 ul) transferred? Approximately how long did it take for “complete lysis” of the cultures – how was “complete lysis” determined? How many viral generations occurred between transfers?

>> A set volume of 50ul of lysate was filtered and transferred to the next round of infection. We thank the reviewer for noticing the phrase “Complete lysis”, which appears in the legend for figure 1. It is incorrect, and in disagreement with the description of the experiment in the main text, where we state: “... remaining cells were removed from phage lysates prior to initiation of the next round of infection“. In each round of infection, lysates were filtered once the decline of the host population ceased. The time period between the infection of bacterial cultures and filtration ranged from one to 3.5 weeks, as different hosts are infected in different efficiencies by the phage. We agree that this information should be presented in the manuscript

and have added these details to the methods section to avoid exceeding the word limit in the main text. However, we do not specify how many viral generations occurred between transfers, as the lysates were not titered in each round and the number of infection cycles completed within each transfer was not determined. That would have required constant monitoring of host populations during the experiment as this changed during adaptive evolution. This would have been extremely labor intensive, perhaps prohibitively so, and very difficult to determine accurately.

5. Line 91: The GenBank accession numbers of the S-TIM4 and Syn19 genome sequences need to be included. If the S-TIM4 genome was sequenced as part of this study, then the methods for assembly and annotation should be included. In addition, all of the sequence reads from each of the evolved populations should be deposited in a public database and the information on how to access those data needs to be included in the manuscript.

>> We agree that this important information should be included in the text. S-TIM4 was not sequenced as part of this study, however, the genome sequence was not submitted to NCBI previously. We have now submitted this genome with all the relevant information to NCBI genome database. The sequence reads of the ancestral and evolved populations are now deposited in NCBI SRA database. All accession numbers are detailed in the supplementary methods section.

6. Genomic Data Analysis: Please clarify “Minimal coverage of 10 reads/strand was required to include polymorphisms in software output”. Is this 10 reads/strands in all, or 10 reads/strand of the variant polymorphism? What was the average coverage of the genome for each of the sequenced populations? Even by sequencing two libraries, a >1% threshold for polymorphisms seems low – particularly if the overall coverage is low. Looking at the list of polymorphisms, there are quite a few below 10%, especially for populations evolved on WH8102. Even if these SNPs are not sequencing errors, how important are SNPs at very low frequency likely to be in influencing overall infection efficiency of the population? More weight should be put on polymorphisms that have greatly increased in frequency in the population and thus are more likely to be under selection. These polymorphisms will also contribute more to the increased infectivity of the viral populations. (for example, a recent article analyzing E coli LTEE populations used a 10% threshold for mutations: Good et al. Nature 2017).

These are important points, and are now also addressed in the methods section. The average coverage of the sequencing libraries is 2000-fold. The 10 read/strand cut-off refers to number of reads containing a specific polymorphism variant on each of the strands. This cut off was used for both strands to make sure that no variant is called erroneously as a result of a strand bias (i.e, only appears on one of the strands).

The error rates of the illumina Hlseq technology are estimated to be 0.001-0.005 substitutions per base, and lower for indels (for example, see “Illumina error profiles: resolving fine-scale variation in metagenomic sequencing data”, Schirmer et al., BMC Bioinformatics, 2016). Our approach of using two technical repeats (i.e, sequencing libraries), for each evolved population allows us to significantly reduce the theoretical error rate that would erroneously be identified as a mutation, and to overcome

potential library preparation errors (see “The Role of Replicates for Error Mitigation in Next-Generation Sequencing”, Robasky et al., Nature Reviews Genetics, 2014). Therefore, the 1% threshold represents quite a conservative approach, taking into account only mutations with frequency much higher than the sequencing error rate.

We agree that mutations with higher frequency hold a higher potential to affect the infection efficiency of the population. However, our aim in this study was not to determine the influence of specific mutations on infection efficiency, but rather to estimate how adaptation to different hosts influences the diversification of phage populations. In order to determine this, it is essential to include these low-frequency mutations, as diversification in practice means emergence of multiple genotypes, even if only in a small portion of the population.

We would like to point out that while Good et al. (2017) only considered mutations with frequencies above 10%, they used a minimal mean coverage per library of just 5-fold, which is much lower than the coverage of libraries constructed in this project. Therefore, our approach is more sensitive and allows the identification of mutations with lower frequencies.

7. Line 100, Figure 4: It would be helpful to standardize how genes are being referred to in the manuscript. In some cases, the ORF number is used, in some cases a gp# is used and in others, the name of the protein. The relationship among these is not clear unless you're looking at TableS3. In Fig.4 – all three designations are used, which is confusing. I would always include the ORF number and then the gp# or protein name if it provides some important additional information.

We have now standardized the reference to genes, defined by their ORF number. In figure 4a we use ORF numbers and gene names, if relevant. In figure 4c, where we discuss actual location of proteins in the virion, we use gene names, which are more informative.

8. Line 120 and Methods: In the methods section, it states that in addition to S-TIM4, Syn19 was also evolved on 5 cultures of each of two hosts (WH8109 and WH8102). However, the only mention of these results is in line 120: “These populations accumulated 4-6 mutations per population”. No data are shown. The inclusion of the results for the evolution of Syn19 in these two hosts would strengthen the paper and provide an interesting comparison for the results obtained with S-TIM4. It would be useful to know if the results obtained for Syn19 show similar trends to S-TIM4. If the Syn19 results are not going to be discussed, then the method section should be revised.

>> We agree that the data for the evolution of Syn19 should be presented, when relevant to this paper. Our primary purpose in including these data in the manuscript was as a control to address the possibility that a significantly higher degree of mutations occurred during evolution of S-TIM4 on *Synechococcus* WH8102 due to something intrinsic in this cyanobacterial strain. As can be seen from the results with Syn19, very few mutations were found when evolved on this cyanobacterial strain, indicating that the rapid adaptation of S-TIM4 on this host is due to the specific host-

phage interaction and not the host per se. We have now included this data in Supplementary table S5, specifying the mutations identified in each of the evolved Syn19 populations on WH8102. In response to this comment we now also discuss these results on lines 131-133, which describes the types of mutations found in the Syn19 populations. It reads: "Examination of the mutation types in Syn19 populations revealed that all were non-synonymous SNP's, further emphasizing the role of positive selection in the adaptive process of cyanophage populations."

9: Line 228: The result of 15 synonymous mutations in one 120 bp segment that optimizes codon usage is very interesting. It is more striking that all of these mutations occur together on the same sequencing reads (Fig S1). It seems very unlikely that through the process of random mutation during 15 rounds of evolution, 15 synonymous mutations in one 120 bp segment would occur all in the same isolates. If these reads are not contamination, perhaps there has been recombination of this segment with another repeated segment elsewhere in the genome? The authors should propose/explain how these mutations likely arose and rule out possible contamination.

>> We thank the reviewer for this suggestion. While we did not find a single region that is identical to the entire mutated 120 bp region, we identified a region identical to the region that contains the upstream 10 synonymous mutations (genomic positions 100242-100308), and a second region identical to the region containing the downstream 5 synonymous mutations (genomic position 10078-10098). Therefore, we can speculate that the emergence of the codon optimizing synonymous mutations could have been the result of recombination events within this gene. We now discuss these findings in the text (lines 245-250), and also changed the legend of Supplementary figure s1 accordingly. The relevant part in the main text now reads: "A comparison of the mutation rich region to the rest of the S-TIM4 genome revealed that the genomic region containing the 10 most upstream mutations is identical to a region downstream to the mutated region, while the region containing the downstream mutations is identical to a region upstream to the mutated region. Therefore, we hypothesize that these polymorphisms are the result of at least two recombination events."

In regards to these results being the outcome of potential contamination - we are confident that these findings are not the result of contaminant, as we did not find any other evidence for possible contaminations elsewhere in the genome. Moreover, assembly of the sequencing reads, as described in the methods section, did not yield contigs which do not belong to the S-TIM4 genome, further reducing the probability of this sample being contaminated.

10: Line 235: Does S-TIM4 carry any tRNAs? If so, which ones?

>> S-TIM4 carries 3 tRNA genes. However, they do not fully complement the optimized codons. We have added the information in the revised manuscript and discuss it in the main text (lines 238-241). Additionally, details of the tRNA genes are available in the annotation file submitted to NCBI.

11. Line 264: This statement should be qualified –for example, in “all cyanomyophages” analyzed in ref. 37. As more cyanomyophages are sequenced, what is considered the “core” genome will change.

>> This is a very relevant remark and is now addressed in the main text. We added the number of phage genomes used to define the “core” and “flexible” genomes (line 281).

12. Lines 270-280: As stated in comment 1, the results of this study could be very informative in trying to understand genomic diversity in natural viral populations. So again, the results should be put in context and compared to what is already known about cyanomyoviral genomic diversity in natural populations. In this experimental setup, the mutations contributing to host adaptation are “not preferentially mediated through mutations in hypervariable regions”. In natural communities, however, cyanophages probably adapt to new hosts not just by mutations, but also through homologous or non-homologous recombination with other related phages or even hosts. In this study, there was only one phage genotype present, therefore allelic recombination, or the acquisition of new genes, was not possible. A sentence should be added to this section acknowledging the limitations of the presented results. In addition, could the proposed model also incorporate, for example, the acquisition of a new AMG that might increase adaptation to a particular host - to give the model a more holistic picture of phage adaption?

>> We agree that these results should be put in context with existing data on cyanomyovirus genomic diversity. This has been done as described in the reply to comment 1.

We agree that our experimental design did not allow us to detect recombination events with other phages as only a single phage was used. However, we would have detected recombination events with hosts, had they occurred. We have now added a sentence to mention the limitation of this experiment with respect to recombination amongst phages. It now reads (line 298): “It should be noted that in natural habitats, cyanophage diversification also occurs through allelic exchange between phages (Gregory et al. BMC Genomics 2016, Marston & Martiny EM 2016), which could not be investigated using our experimental system as only a single phage was present”.

While it would be interesting to update the model in the way the reviewer suggests, we do not think it is appropriate to do so since, based on our current observations, we did not observe the acquisition of new genes. Thus, we’d need to perform more long-term experiments to see if such gene acquisitions result before we feel we could update the model in this way. Clearly, such longer-term experiments are beyond the scope of our current study.

Answers to Reviewer #3 comments:

The paper is an interesting contribution by a well known research group that seeks to examine how diversification in viral genomes occurs through the

infection of hosts with different efficiencies. In general I like the idea of the study but I have a few question.

The authors use the term "sub-optimal" but never really define this in a quantitative sense. Can they do this? It would seem to me that a qualitative definition (which is how I read the paper) is not as strong.

>> We agree that a more quantitative definition would, in principle, be stronger and more convincing. However, since optimality acts along a continuum, with some hosts more optimal than others, a quantitative definition is problematic. As stated in the manuscript we consider the most "optimal" host to be the one that is infected most efficiently. Thus, our definition is actually a relative one which seems most appropriate for the phenomenon we are discussing in the manuscript.

Page three. "In natural habitats, the abundance of cyanobacterial hosts...". These are a mix of hosts and non-hosts, right? Or putative hosts as in some case the physiological conditions / environmental conditions need to be right to make things work.

>> Yes we agree. We had initially intended to refer to particular cyanobacterial types that would serve as hosts for certain phage. Nevertheless, we understand that this sentence was somewhat misleading and have changed it accordingly. It now reads (line 51): "In natural habitats, the abundance of cyanobacterial types often changes over space and time¹⁸⁻²¹, as a result of ...".

Line 54 and elsewhere. The authors use "in order to" a few times in the paper and they can delete almost every one of these.

>> We have now deleted all of these.

Line 55. Would forming a lysogen not also be a solution to this problem?

>> Lysogeny, in principal, could be a solution for this problem. However, in marine T4-like cyanophages, lysogeny is not an option. These phages do not encode integrase genes and there is no evidence for lysogeny in this family of cyanophages.

Line 68. Optimality refers to efficiency. You need to define efficiency here. Is this rate of contact success? Rate of infection of individuals? Fitness (particles produced per infection, or infectious particles produced per infection). I am trying to illustrate that different readers could take this different ways and you need to bring them along with you on this.

>> We agree that "efficiency" can be interpreted differently by different readers. To avoid this we clarify the term, defining efficiency as the rate of decline of host populations following viral infection. This measure encompasses all levels of change in optimality, from increased adsorption, reduced length of the latent period and increased burst size, and is one of the reasons it was used. The text now clarified this and reads (lines 69-73): "...efficiency with which the phages infect their

cyanobacterial hosts, measured by the rate of decline of the host population. This has been shown previously to be directly correlated with phage fitness and encompasses multiple ways in which fitness can change, including rate of adsorption, length of the latent period and burst size”.

Line 89, I am left wondering about "wild-type"...presumably this has been in someone's lab for many generations? Are there effects from previous rounds of culture that could influence your observations (same with the hosts for that matter).

>> S-TIM4 underwent minimal rounds of culturing in the lab. However, it is indeed possible that this phage has changed during the isolation and culturing process. Therefore, we have changed “wild-type” to “ancestral” throughout the manuscript.

Line 94. I do not see how supplemental table 1 shows efficiency. I see fluorescence readings. Is efficiency defined as the amount of a population killed (if so it would be nice to back this up with cell counts)

>> We thank the reviewer for this comment. Efficiency is now defined as the rate of decline of host populations following viral infection (see reply to the comment regarding line 68 in the original manuscript). The title of supplemental table 1 was misleading. The table denotes cell density from chlorophyll a fluorescence readings, 10 days post infection, relative to that in the uninfected control cultures. We have changed the titles accordingly.

The use of chlorophyll a fluorescence as a proxy for cell density is a standard procedure in the study of cyanobacteria (Moore et al. 2007), and is equivalent to the use of optical density in heterotrophic bacterial cultures. When cultures are grown at the same light intensity for the duration of the experiments (as done in our study), these fluorescence readings are directly correlated with cell density. We have now cited Moore et al. (2007) for this in the Methods section of the manuscript.

Line 107. Is it possible the hosts just have better proof-reading or repair mechanisms? How would that play into your observations. I realize you mention this a little later on this page but the mention is very brief and I am not sure how comparable these two viruses are. Just playing devil's advocate here.

The possibility of more relaxed proof reading or repair mechanism in the *Synechococcus* host was indeed something we considered. Therefore, we addressed this possibility by evaluating the evolution of a second phage, Syn19, when evolved on the same *Synechococcus* host strain. For this phage only a few mutations were identified, indicating that the high number of mutations found for S-TIM4 when evolved on this strain was not due to some intrinsic difference in this host relative to the *Prochlorococcus* hosts. It should be noted that Syn19 and S-TIM4 are both cyanophages from the *Myoviridae* family, having similar morphology, life-style, GC-content, genome length, gene content and a similar number of open reading frames.

Line 149. I am not sure what "clustered most differently" means....please clarify

>> We agree with the reviewer that this was not clear and have changed this sentence to better clarify our meaning. It now reads (lines 158-159): “ ... were clustered together to form the group with the highest distance from the ancestral population”.

Line 211. The authors keep mentioning positive selection. But I am not sure what negative selection would look like...I think that needs to be clarified.

>> Strong negative selection would result in an enrichment of synonymous SNP's and genotypic convergence of the ancestral and evolved populations. We now clarify this point in the part that relates to figure 2a, in which negative and positive selection are first mentioned (lines 109-110). It now reads: “The fraction of non-synonymous mutations within coding regions is expected to be ~75% in the absence of selection and <75% if negative selection is the dominant type of selection²⁷”.

Line 242. Is this DUF region also found in hosts?

>> This region is not found on any of the hosts used in this work. We added this information to line 259, which reads: “This gene, which has no homologs in the bacterial strains used in this study, was...”

Line 256. Confusing heading. Occurs in conserved and variable regions....so everywhere? Random?

>> We changed the heading to be less confusing. It now reads: “Phage adaptation does not occur through preferentially mutating variable genomic regions”. We would like to clarify that the positive selection of mutations is not random, however it is simply not determined by the conservation level of the genomic regions.

Line 287. again highest efficiency? So lysis? Fewer resistant cells?

As mentioned in the reply to the comment concerning line 68, efficiency is now defined in the text as the rate in which the host population declines and is correlated with phage fitness.

Line 309. Matteson et al. (FEMS MICROB ECOL 2013) show that when Synechococcus massively decreases in the water column the viruses massively decrease as well in the days that follow...This supports the first part of your final statement (alas, they do not appear to have done the sequencing that would really prove your point).

>> We thank the reviewer for this suggestion. The publication by Matteson et al. indeed supports the first part of this statement, and is now cited.

Reviewers' comments:

Reviewer #1 (Remarks to the Author):

I was a pleasure to read this paper and I enjoyed the revised version of the manuscript. All of my comments on the previous version of the manuscript were now appropriately addressed.

Reviewer #2 (Remarks to the Author):

I would like to thank the authors for their detailed responses to my initial round of comments. The manuscript has been revised and most of my questions and concerns have been adequately addressed.

Two remaining concerns:

1. Lines 71-73: Here Turner et al 2012 is cited to suggest that host lysis rate is a good proxy for viral infection. The added text is a bit misleading. It sounds as though "It has been shown previously" in this cyanobacterial/cyanophage system - when in fact, Turner et al were using a heterotrophic host and RNA virus. The text should indicate this: "It has been shown with a heterotrophic host and RNA virus that host decline can be directly correlated to phage fitness.....". I still think that it is a bit worrisome that the authors haven't demonstrated this relationship for their system - especially since many of the conclusions are based on this proxy.
2. In Figure 3 (all parts) "wt" needs to be changed to "Anc" to match the figure legend and text.

Reviewer #3 (Remarks to the Author):

Thank you for your efforts in responding to my previous comments. A job well done.

Answers to Reviewers comments:

Reviewer #1

I was a pleasure to read this paper and I enjoyed the revised version of the manuscript. All of my comments on the previous version of the manuscript were now appropriately addressed.

>> Thank you very much.

Reviewer #2

I would like to thank the authors for their detailed responses to my initial round of comments. The manuscript has been revised and most of my questions and concerns have been adequately addressed.

Two remaining concerns:

1. Lines 71-73: Here Turner et al 2012 is cited to suggest that host lysis rate is a good proxy for viral infection. The added text is a bit misleading. It sounds as though "It has been shown previously" in this cyanobacterial/cyanophage system - when in fact, Turner et al were using a heterotrophic host and RNA virus. The text should indicate this: "It has been shown with a heterotrophic

host and RNA virus that host decline can be directly correlated to phage fitness.....". I still think that it is a bit worrisome that the authors haven't demonstrated this relationship for their system - especially since many of the conclusions are based on this proxy.

>> We understand the concern of the reviewer and editor as to whether the findings in Turner et al. 2012 are generalizable and would therefore also hold true for our system. To address this we have turned to previously published results by Schwartz & Lindell 2017 (published in *The ISME Journal*) in which fitness was directly measured for variants of dsDNA P-SSP7 phage variants as was the time to lysis of two *Prochlorococcus* MED4 host strains during infection with these variant phages. The fitness data are presented in Fig. 2a and the time to lysis data are presented in Fig. S1 of Schwartz & Lindell 2017. We used this data to compare the observed time to lysis and fitness of 52 phage-host interactions and show that the two measurements are negatively correlated, i.e. higher fitness results in a shorter time to complete lysis (see new Supplementary Fig. 1a). This is essentially the same as the experimental results reported by Turner et al. 2012. We further compared fitness with the density of the infected cultures (from chlorophyll *a* fluorescence measurements) at 6 days after infection. Again, these two measurements show a significant negative correlation (see new Supplementary Fig. 1b) and are directly comparable to the analyses we use in our current manuscript. Thus, these findings support the use of host density at a given time point after infection as an indication of the fitness of the viral population in a closely related cyanobacterial-cyanophage system. It is relevant to note that the same wild-type *Prochlorococcus* MED4 strain is used in both the Schwartz & Lindell 2017 study and in our current study. We now present these comparisons and correlations in Supplementary Figure 1 of the manuscript and cite Schwartz & Lindell 2017 for the data.

To summarize, the additional analyses added to the manuscript indicate that there is a negative correlation between the time to lysis of the host and the fitness of the phage in a marine cyanobacterium and phage system. Thus, the same significant trend was observed in two completely different systems: a heterotrophic host and RNA phage in Turner et al. 2012, and a marine cyanobacterium with a dsDNA phage here. This strongly supports that a negative correlation between fitness and time to lysis is a general phenomenon that spans vastly different host-phage systems. We

hope that this new analysis now alleviates the reviewer's and editor's concern about using host density at a set time after infection as proxy for phage fitness.

We now present this in the text on lines 71-75 by stating: "This has been shown previously to be directly correlated with phage fitness using a heterotrophic host and an RNA phage²⁶ and a marine cyanobacterial host and a DNA cyanophage²⁷ (see Supplementary Figure 1). This measure encompasses multiple ways in which fitness can change, including rate of adsorption, the latent period (the time at which new phage progeny are released) and burst size."

We also add a reference to these findings from two other places in the manuscript. First, in the text prior to the presentation of the relevant phenotypic results. Lines 146-148 now read: "Next, we sought to determine whether S-TIM4 populations that had evolved different genotypes had distinct phenotypes, as reflected by their infectivity profiles which are an indication of phage fitness²⁶ (Supplementary Figure 1)." Second in the legend of Figure 3 (line 521) we added a reference to Supplementary Figure 1 at the end of the following sentence: "Lower values indicate more rapid declines in the infected host population, i.e. increased fitness (Supplementary Figure 1)."

2. In Figure 3 (all parts) "wt" needs to be changed to "Anc" to match the figure legend and text.

>> Thank you for noticing this mistake. We have now corrected figure 3 accordingly.

Reviewer #3

Thank you for your efforts in responding to my previous comments. A job well done.

>> Thank you very much.

REVIEWERS' COMMENTS:

Reviewer #2 (Remarks to the Author):

Thank you for addressing my concern about the relationship between host decline and viral fitness by including additional analyses that indicate a negative correlation between host lysis and viral fitness in this system (Supplementary Figure 1). All of my comments and concerns have now been addressed.